# Functional Attention:
# From Pairwise Affinities to Functional Correspondences

**Jiefang Xiao** [1 2]  **Maolin Gao** [1 2]  **Simon Weber** [3]  **Guandao Yang** [4]  **Daniel Cremers** [1 2]

## Abstract

Learning mappings between infinite-dimensional function spaces, or operator learning, is essential for many machine learning applications. Although transformer-based operators are popular, they often rely on token-wise attention. These methods treat continuous fields as discrete tokens and usually ignore the global functional structure. We introduce *Functional Attention*, which reinterprets attention as a functional correspondence between adaptive bases. Inspired by geometric functional maps, our method replaces softmax affinities with structured linear operators. This yields a compact, generalizable, resolution-invariant representation that explicitly captures global dependencies. Experiments demonstrate that *Functional Attention* can match state-of-the-art performance in many operator learning tasks, including solving PDEs, 3D segmentation, and regression, while remaining robust to varying discretizations. Project page is available at https://github.com/xjffff/FUNCATTN.

## 1. Introduction

Many machine learning problems can be formulated as learning a mapping between infinite-dimensional function spaces, a task also known as operator learning. The operator learning paradigm has a significant impact on numerous applications, such as solving partial differential equations (PDEs), computational designs by inverting PDEs, or physical simulations (Kovachki et al., 2023). While the mainstream machine learning community has moved toward deep learning for most tasks, building a neural network architecture suitable for operator learning is challenging, largely due to the difficulties in representing continuous functions.

Most of the first neural operator architectures try to take advantage of functional bases to represent and process functions. For example, the Fourier Neural Operator (FNO) (Li et al., 2021), the Multiwavelet-based Model (Gupta et al., 2021a) and U-NO (Rahman et al., 2023) demonstrated that learning in the spectral domain allows building efficient mapping between physical fields, enabling fast PDE solving. Other than the Fourier basis, there are other network architectures exploring Laplacian eigenbasis (Sharp et al., 2022). While these heuristically decided bases can be effective in tasks where the basis choice is suitable, these architectures can potentially be limited in their representational powers, largely due to the inductive bias in their design. It is unclear how such hand-picked bases can scale to wider applications.

Moreover, the community has witnessed the power of the transformer-based architectures, which have achieved SOTA in many tasks in NLP, vision and operator learning (Vaswani et al., 2017; Dosovitskiy, 2020; Wu et al., 2024). Despite its success, these attention-based networks represent a function using a discrete set of tokens. There is little discussion in connecting such a token-centric perspective with the operator learning setup. Moreover, these methods, representing a function using a collection of its samples, (1) may scale poorly, since the scaled dot-product attention mechanism requires quadratic computation with respect to the number of samples needed to represent the function, (2) ignore the global functional structure, leading to redundant parameterization and (3) miss a principled way to maintain consistency across different resolutions or irregular meshes.

In this work, we propose an alternative perspective of the attention mechanism for operator learning. Rather than viewing attention as a mechanism for computing pointwise correspondences between tokens, we reinterpret it as a *functional correspondence* between learned function spaces. Our approach draws inspiration from the seminal *functional maps* framework in geometry processing (Ovsjanikov et al., 2012), where correspondences between complex 3D nonrigid shapes are represented as simple linear operators acting on functional bases. This alternative perspective allows us

[1]Technical University of Munich, Germany [2]Munich Center for Machine Learning (MCML), Germany [3]PIXL, Department of Computer Science, University of Oxford, United Kingdom [4]ECE, University of Texas at Austin, USA. Correspondence to: Maolin Gao <maolin.gao@tum.de>, Simon Weber <simon.weber@cs.ox.ac.uk>.

*Proceedings of the 43rd International Conference on Machine Learning*, Seoul, South Korea. PMLR 306, 2026. Copyright 2026 by the author(s).

to design an attention formalism, which can capture intrinsic structural properties of the underlying problem. Our framework, which we call *Functional Attention (*FUNCATTN*)*, provides a unified and theoretically grounded approach to operator learning.

In summary, our contribution is two-fold: First, we introduce a novel attention paradigm based on functional correspondences, establishing a principled connection between the standard attention formulation and the functional maps framework. Second, we demonstrate that this perspective unlocks a new design space for attention mechanisms, and we present an effective instantiation - FUNCATTN, that is versatile across a diverse set of tasks, including PDE solving, 3D segmentation, and regression. Across these settings, FUNCATTN consistently achieves state-of-the-art performance and exhibits strong robustness, generalizing reliably across datasets and resolutions.

## 2. Related Work

Recent research on operator learning has explored a wide range of architectures to balance expressiveness, computational efficiency, and geometric flexibility, based on attention networks and their derivatives. In our literature review, we focus on neural operator methods and attention mechanisms that model mappings between infinite-dimensional function spaces. These works form the conceptual basis for reinterpreting attention as a functional operator, which directly motivates our spectral formulation.

### 2.1. Attention and its Derivatives

Standard scaled dot-product attention (Vaswani et al., 2017) is formulated as:

$$\text{Attention}(\mathbf{Q}, \mathbf{K}, \mathbf{V}) = \text{Softmax}\left(\frac{\mathbf{Q}\mathbf{K}^\top}{\sqrt{d_k}}\right)\mathbf{V} \quad (1)$$

where $\mathbf{Q}, \mathbf{K}, \mathbf{V}$ represent the query, key, and value matrices, respectively. While powerful, this mechanism suffers from quadratic complexity with respect to the context length, posing a significant challenge for scaling.

Linear attention (Katharopoulos et al., 2020) addresses this by introducing kernel functions $\phi(\cdot)$ that enable matrix associativity, allowing the computation to be reordered as $\phi(\mathbf{Q})(\phi(\mathbf{K})^\top \mathbf{V})$. This reduces complexity from quadratic to linear. Various kernel designs have been proposed, including random Fourier features (Peng et al., 2021), positive random features in Performer (Choromanski et al., 2020), cosine reweighting (Qin et al., 2022), MLP-based maps (Zhang et al., 2024), and discrete cosine transforms (Chen et al., 2024). However, these methods operate in the token space and overlook intrinsic geometric and physical structures.

Our work differs from this by introducing a functional perspective that incorporates these structures, which we demonstrate to be more effective empirically.

**Low-rank approximation methods** seek to compress attention representations to reduce cost. Linformer (Wang et al., 2020) projects keys and values into a low-dimensional space, while Nyströmformer (Xiong et al., 2021) utilizes the Nyström method for softmax approximation. Other variants include Perceiver's (Jaegle et al., 2021) fixed learnable queries, Reformer's (Kitaev et al., 2020) locality-sensitive hashing, and Monarch Attention's (Yaras et al., 2025) structured matrix constraints. Unlike these approaches, which focus on approximating the standard attention matrix, our method introduces a novel formulation that performs least-squares regression in a learned spectral space, offering a preferable alternative for capturing complex dependencies.

### 2.2. Neural Operator

Neural operators learn mappings between infinite-dim function spaces to provide mesh-free approximations of PDE solution operators. The Fourier Neural Operator (FNO) (Li et al., 2021) parameterizes integral kernels in the Fourier domain via the FFT, achieving efficient $O(n \log n)$ complexity. However, FNO is restricted to uniform Cartesian grids and suffers from periodic boundary assumptions. While Geo-FNO (Li et al., 2023a) uses diffeomorphisms to handle irregular domains, it remains dependent on global charts that are difficult to construct for complex topologies. In contrast, our approach draws inspiration from spectral transformations, but designs attention directly in the spectral space. Unlike FNO-based methods, our framework is not limited to grid-based PDEs and generalizes to broader learning tasks such as regression and segmentation.

To accommodate arbitrary geometries, graph-based methods like GNO (Li et al., 2020), GINO (Li et al., 2023b), and UPT (Alkin et al., 2024) utilize message passing or latent super-nodes. Alternatively, Transformer-based architectures such as OFormer (Li et al., 2023c), GNOT (Hao et al., 2023), and FactFormer (Li et al., 2023d) leverage attention to handle geometry flexibly. The Galerkin Transformer (Cao, 2021) interprets linear attention as a Petrov-Galerkin projection, treating columns of $\mathbf{Q}, \mathbf{K}, \mathbf{V}$ as samples of functions in Hilbert spaces. While we also adopt this functional view, we distinguish our work by explicitly learning a set of bases in the query and key-value spaces via a simple feed-forward architecture. This separation of function and basis, inspired by the functional maps framework (Ovsjanikov et al., 2012; Fumero et al., 2024; Behmanesh et al., 2024), offers greater expressiveness than the implicit basis change in Galerkin attention. Furthermore, we compute our attention via an *optimal linear solve* in the spectral domain rather than as an approximation of classical attention.

Recent works like Transolver (Wu et al., 2024) and Transolver++ (Luo et al., 2025) reduce costs by learning intrinsic physical states through a "slice-and-attend" paradigm. Our method generalizes this concept; while their slicing and de-slicing layers are conceptually similar to our spectral transforms, we leverage a more general spectral framework. Our learned functional coefficients generalize their physics-aware tokens to capture intrinsic structures beyond pure physics. Additionally, while Transolver applies standard scaled dot-product attention, our *optimal linear solve* in the spectral space experimentally demonstrates highly competitive performance.

# 3. The Problem & Motivation

In this section, we first revisit the task of operator learning, which defines our learning objective at the level of mappings between function spaces (Section 3.1).Then we discuss the common practice to date, which employs tokenized representations via attention mechanism (Section 3.2). This approach ignores the geometric structure of the underlying problem and is data-inefficient (Section 3.3). This limitation motivates a functional view of the problem inspired by the seminal work of functional maps (Ovsjanikov et al., 2012), briefly introduced as background (Section 3.4).

## 3.1. Operator Learning Formulation

We consider the task of learning mappings between an input and output space. Let $\Omega \subset \mathbb{R}^d$ be a bounded space. Consider $\mathcal{F} = \mathcal{F}(\Omega; \mathbb{R}^{d_f})$ and $\mathcal{G} = \mathcal{G}(\Omega; \mathbb{R}^{d_g})$ be separable Banach spaces of function taking values in $\mathbb{R}^{d_f}$ and $\mathbb{R}^{d_g}$ respectively. We aim to learn the underlying mapping between such functions, that can be formalized as a nonlinear operator: $\mathcal{O} : \mathcal{F} \to \mathcal{G}$.

Suppose we are given a dataset of observed pairs $\{(f_j, g_j)\}_{j=1}^N$ where $f_j \sim \mu$ are i.i.d. samples from a probability measure $\mu$ supported on $\mathcal{F}$, and $\mathcal{O}^*(f_j) = g_j$ denotes the ground truth mapping. We aim to use a neural network to approximate $\mathcal{O}^*$ by $\mathcal{O} : \mathcal{F} \times \theta \to \mathcal{G}$ with learnable parameter $\theta$. This provides us a framework for learning infinite dimensional function through an optimization problem with a cost functional $\mathcal{L} : \mathcal{G} \times \mathcal{G} \to \mathbb{R}$:

$$\min_{\theta \in \Theta} \mathbb{E}_{f \sim \mu} \left[ \mathcal{L} \left( \mathcal{O}(f; \theta), \mathcal{O}^*(f) \right) \right] \tag{2}$$

Many scientific and geometric learning tasks are naturally posed as mappings between *functions* rather than finite-dimensional vectors: the input is a continuous quantity, such as a coefficient field, a forcing or an observation field, and the output is another continuous quantity, such as a solution field, a future state or a reconstructed field. Formulating the problem as an operator learning makes the learning target independent of a particular discretization, enabling resolution-invariant generalization across meshes

or sampling densities (cf. Tab. 5).

## 3.2. Tokenized Representations

While operator learning formalism provides a framework that is free of discretization, practical neural architectures, including attention-based models, operate on finite samples of functions. As a result, these models do not learn operators directly, but rather implement pointwise mappings on discretized evaluations of the input field. For instance, an input function $f \in \mathcal{F}$ is evaluated at locations $\{x_i\}_{i=1}^n$ to obtain $\{f(x_i)\}_{i=1}^n$, which are stacked together to obtain the input token matrix $\mathbf{X} \in \mathbb{R}^{n \times d}$, where $n$ is dubbed as the context length. The token matrix $\mathbf{X}$ is further used to compute query, key and value by $\mathbf{Q} = \mathbf{X}\mathbf{W_Q}, \mathbf{K} = \mathbf{X}\mathbf{W_K}, \mathbf{V} = \mathbf{X}\mathbf{W_V}$, where $\mathbf{W_Q} \in \mathbb{R}^{d \times d_q}, \mathbf{W_K} \in \mathbb{R}^{d \times d_k}, \mathbf{W_V} \in \mathbb{R}^{d \times d_v}$ are learnable weights and $d_q = d_k$. Attention-based architectures have been increasingly employed to leverage their ability to capture long-range dependencies in physical fields or latent space (Cao, 2021; Li et al., 2023c; Hao et al., 2023; Wu et al., 2024). However, standard attention treats each row of $\mathbf{X}$ as an independent token—a design inherited from NLP. As Cao (2021) noted, the columns of $\mathbf{Q}/\mathbf{K}/\mathbf{V}$ matrices can be seen as discretizations of functions in Hilbert spaces. However, they merely showed that theoretically the inner products computed during the Galerkin-type attention step act as the coefficients for a linear combination of learned bases in the value space, but did not specify how these bases and coefficients can be computed in practice. Nevertheless, this intriguing perspective motivates the questions: can this functional view be leveraged to design a data-efficient attention mechanism by considering the underlying structure of problem?

## 3.3. Motivation

Treating each token independently and ignoring their underlying relationship is suboptimal, especially when the problems manifest geometric or physical structures. In standard attention, the dense score matrix that maps values to outputs is represented explicitly as pointwise affinities between discrete samples. While this representation is convenient, it tightly couples the complexity of attention to the number of tokens, and implicitly assumes that meaningful correspondences must be established at the level of individual points. However, in many settings of interest, such as operator learning (Li et al., 2021), physical field modeling (Wu et al., 2024) or dense prediction (Devlin et al., 2019), tokens arise as samples of underlying functions. In these cases, the intrinsic complexity of the signal is often *far lower* than its discretization resolution, and many distinct point-wise affinity matrices induce nearly identical transformations at the function level. This makes the dense, token-level parameterization both computationally inefficient and conceptually redundant. These observations suggest that the

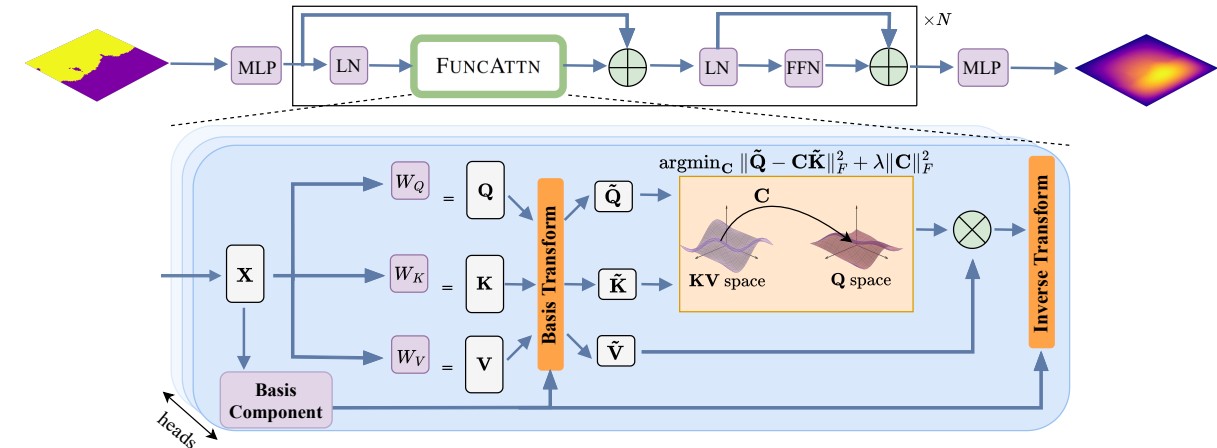

*Figure 1.* **Architecture Overview.** *Top:* Input functions are encoded by MLP, processed through $N$ FUNCATTN blocks, and decoded by MLP. *Bottom:* In each FUNCATTN Module, $\mathbf{Q}, \mathbf{K}, \mathbf{V}$ are transformed to the spectral domain where cross-space attention computes optimal linear mapping $\mathbf{C}$, then inverse-transformed. Purple blocks denote learnable layers. MLP, LN, and FFN stand for Multi-Layer Perceptron, Layer Norm, and Feed-Forward Network, respectively. The Basis Component learns an adaptive basis, and the Basis Transform and Inverse Transform modules apply the learned basis for computing functional attention.

key object of interest in the attention computation is not the pointwise affinity matrix itself, but the linear operator it induces on function spaces. Since attention ultimately defines a map from values to outputs, it is natural to ask whether this operator can be represented directly between function spaces, without resorting to explicit point-to-point correspondences. This leads to a functional perspective on attention: *how should one define attention as a linear operator that transforms a function from the key–value space to the query space, given only discrete samples?* Addressing this question is the central goal of our work.

### 3.4. Functional Maps

To this end, we draw inspiration from the functional maps framework (Ovsjanikov et al., 2012), which provides a principled representation of correspondences between spaces through linear operators acting on function spaces. It is originally proposed in shape matching realm: rather than seeking combinatorially hard point-to-point correspondences between manifolds $\mathcal{M}$ and $\mathcal{N}$, functional maps shift the problem to function spaces $L^2(\mathcal{M})$ and $L^2(\mathcal{N})$. This perspective offers two key advantages: the transformation between function spaces is linear even when the underlying point map is combinatorial, and the correspondence can be compactly represented in $k$ spectral bases, reducing complexity from $O(n^2)$ to $O(k^2)$ where $k \ll n$. Specifically, given $m$ pairs of descriptor function represented in their respective truncated Laplace-Beltrami eigenbases as matrices $\mathbf{A} \in \mathbb{R}^{k_{\mathcal{M}} \times m}$ and $\mathbf{B} \in \mathbb{R}^{k_{\mathcal{N}} \times m}$, the functional map $\mathbf{C} \in \mathbb{R}^{k_{\mathcal{N}} \times k_{\mathcal{M}}}$ between two functional spaces is estimated via regularized least-squares:

$$\mathbf{C} = \arg\min_{\mathbf{C}} \|\mathbf{B} - \mathbf{C}\mathbf{A}\|_F^2 + \lambda \, \mathcal{R}(\mathbf{C}) \qquad (3)$$

This formulation reduces a combinatorial problem to a convex optimization that can often be solved in closed form.

In the following, we adopt this functional viewpoint and show how a novel attention mechanism can be formulated as the estimation of a compact linear operator between learned functional spaces, avoiding explicit pointwise matching. This leads to a Functional Attention (FUNCATTN) that replaces softmax-based pointwise affinities with a basis-aware operator learned through least-squares objectives, closely mirroring the functional maps paradigm while remaining fully compatible with modern attention architectures.

## 4. Functional Attention

Drawing on the theory of functional maps, we introduce a novel attention mechanism that achieves compact, basis-aware transport. We first introduce the overall idea of FUNCATTN. Then we dive into its two key components, namely the estimation of optimal linear transport (Section 4.1) and the basis selection (Section 4.2). See Fig. 1 for the architecture overview. We finally prove the continuity of FUNCATTN (Section 4.3). For an analysis of the computational complexity, we refer the reader to Appendix B.

**Main Idea.** We design FUNCATTN by asking: *what linear operator $\mathcal{T} : \mathcal{F}(\mathcal{X}) \to \mathcal{F}(\mathcal{Y})$ best explains the transport from the key-value space to the query space?* If we equip both function spaces with bases $\{\boldsymbol{\psi}_j\}_{j=1}^k$ and $\{\boldsymbol{\phi}_i\}_{i=1}^k$, then $\mathcal{T}$ admits a matrix representation $\mathbf{C} \in \mathbb{R}^{k \times k}$. The correspondence problem reduces from estimating an $n \times n$ affinity matrix to estimating a compact $k \times k$ operator.

Concretely, let $\boldsymbol{\Phi} \in \mathbb{R}^{n \times k}$ and $\boldsymbol{\Psi} \in \mathbb{R}^{n \times k}$ denote bases at the query and key-value space respectively, where each column represents a basis function evaluated at the $n$ discretization points. The spectral coefficients of queries and

keys are:

$$\tilde{\mathbf{Q}} = \mathbf{\Phi}^{\dagger}\mathbf{Q} \in \mathbb{R}^{k \times d}, \quad \tilde{\mathbf{K}} = \mathbf{\Psi}^{\dagger}\mathbf{K} \in \mathbb{R}^{k \times d} \quad (4)$$

where $\mathbf{\Phi}^{\dagger} = (\mathbf{\Phi}^{\top}\mathbf{\Phi})^{-1}\mathbf{\Phi}^{\top}$ denotes the Moore–Penrose pseudo-inverse, similarly for $\mathbf{\Psi}^{\dagger}$. The functional attention operator $\mathbf{C}$ is the underlying transport defined by $\mathbf{K}$ and $\mathbf{Q}$ in the spectral space, which can be deployed to transport $\mathbf{V}$.

$$\text{FUNCATTN}(\mathbf{Q}, \mathbf{K}, \mathbf{V}) = \mathbf{\Phi}\,\mathbf{C}\,\tilde{\mathbf{V}} \quad (5)$$

where $\tilde{\mathbf{V}} = \mathbf{\Psi}^{\dagger}\mathbf{V}$ are the spectral coefficients of values. This yields a compact transport mechanism that is continuous, basis-aware, and as shown later, naturally stable under regularization.

*Remark* 4.1. While Eq. (4) prescribes the Moore–Penrose pseudo-inverse as the canonical projection onto $\text{span}(\mathbf{\Phi})$, in practice we use the transpose $\mathbf{\Phi}^{\top}$ instead, due to its superior numerical stability and runtime complexity, and analogously for $\mathbf{\Psi}$. The two coincide when $\mathbf{\Phi}$ is orthonormal; in the general case, $\mathbf{\Phi}^{\top}\mathbf{Q}$ returns the inner products $\langle \mathbf{\Phi}_{:,j}, \mathbf{Q} \rangle$ for $j = 1, \ldots, k$, which form a legitimate function space representation of $\mathbf{Q}$. Please refer to Appendix D.2 for a detailed discussion.

## 4.1. Estimating the Operator C

Inspired by functional maps, we formulate it as a Tikhonov-regularized least-squares problem: find $\mathbf{C}$ that minimizes the reconstruction error between query and transported key in the spectral domain,

$$\min_{\mathbf{C}} \ \|\tilde{\mathbf{Q}} - \mathbf{C}\tilde{\mathbf{K}}\|_F^2 + \lambda\|\mathbf{C}\|_F^2 \quad (6)$$

where $\lambda > 0$ controls regularization strength, Setting the gradient to zero yields the closed-form solution:

$$\mathbf{C}^* = \tilde{\mathbf{Q}}\tilde{\mathbf{K}}^{\top}\big(\tilde{\mathbf{K}}\tilde{\mathbf{K}}^{\top} + \lambda\mathbf{I}_k\big)^{-1} \quad (7)$$

Substituting into (5) gives the complete functional attention mechanism:

$$\text{FUNCATTN}(\mathbf{Q}, \mathbf{K}, \mathbf{V}) = \mathbf{\Phi}\left[\tilde{\mathbf{Q}}\tilde{\mathbf{K}}^{\top}\big(\tilde{\mathbf{K}}\tilde{\mathbf{K}}^{\top} + \lambda\mathbf{I}_k\big)^{-1}\right]\tilde{\mathbf{V}} \quad (8)$$

Beyond computational savings, the compact $k \times k$ operator acts as an implicit low-rank constraint on the attention mechanism, which can improve generalization on structured data.

*Remark* 4.2. The Tikhonov term $\lambda\|\mathbf{C}\|_F^2$ is introduced for numerical stabilization of the linear solve in Eq. (7). We provide an empirical sensitivity analysis of $\lambda$ and the resulting condition number in Appendix D.3.

## 4.2. Choice of Basis

The basis matrices determine how input features are projected into spectral coefficients and how the functional attention operator $\mathbf{C}$ captures correspondences in the compressed space.

**Fixed Spectral Basis.** Classical approaches use predetermined bases such as Fourier bases (Li et al., 2021). It is computationally via fast transforms, but fixed bases assume a regular grid structure, which may not align with task-specific features in the data, hence limiting its expressiveness.

**Learned Adaptive Basis.** To address this limitation, we learn bases that adapt to the input data. Inspired by (Wu et al., 2024), we construct data-dependent basis functions. Specifically, given input $\mathbf{X} \in \mathbb{R}^{n \times d}$, the basis are estimated as following:

$$\mathcal{B} = \text{Softmax}\big(\text{Linear}(\mathbf{X})\big) \in \mathbb{R}^{n \times k} \quad (9)$$

where $\mathcal{B}$ is $\mathbf{\Phi}$ (resp. $\mathbf{\Psi}$) for query (resp. key-value) space, and $\text{Linear} : \mathbb{R}^d \to \mathbb{R}^k$ is a fully connected layer and $\text{Softmax}()$ operation is applied along the $k$ dimension. We interpret the learned bases as a generalization of classical piecewise-constant ($P_0$) elements, as stated in the following proposition.

**Proposition 4.3** (Learnable Basis as Generalized $P_0$ Elements)**.** *Define the soft basis functions via a score function* $s : \Omega \to \mathbb{R}^k$ *for a point* $x \in \Omega$ *and any* $j \in \{1, \cdots, k\}$:

$$\phi_j(x; \tau) = \frac{\exp(s_j(x)/\tau)}{\sum_{l=1}^{k} \exp(s_l(x)/\tau)}, \quad \tau > 0 \quad (10)$$

*Then: (i)* $\{\phi_j\}$ *satisfies the partition-of-unity property* $\sum_j \phi_j(x; \tau) = 1$ *for all* $\tau$; *(ii) as* $\tau \to 0$, $\phi_j(x; \tau) \to \mathbf{1}_{\Lambda_j}(x)$ *where* $\Lambda_j = \{x : s_j(x) > s_l(x), \forall l \neq j\}$, *recovering classical* $P_0$ *piecewise constant elements.*

A proof is provided in Appendix A.1. This formulation offers two advantages over fixed bases: *(i)* the partition geometry adapts to each input, which allows for capturing intrinsic structure such as semantic, geometric, and physical information (Wu et al., 2024); *(ii)* the softmax normalization ensures that the weights remain bounded and sum to one, which prevents degenerate solutions. We further show that Functional Attention is equivalent to a learnable integral operator on $\Omega$ (proof in Appendix A.2).

*Remark* 4.4 (General Basis). Unlike spectral methods that impose orthogonality or frequency alignment, the learned basis by Eq. (9) is unconstrained. This low-bias design empirically yields expressive representations (cf. Table 7).

## 4.3. Continuity of Functional Attention

Sections 4.1 and 4.2 introduced the two ingredients of FUN-CATTN: the Tikhonov-regularized operator $\mathbf{C}$ in Eq. (7) and the softmax basis $\mathbf{\Phi}, \mathbf{\Psi}$ in Eq. (9), parameterized respectively by $\mathbf{W}_{\mathbf{\Phi}}, \mathbf{W}_{\mathbf{\Psi}} \in \mathbb{R}^{d \times k}$. We now show that the combination of these two ingredients yields a layer whose Lipschitz constant is controlled by the regularization parameter $\lambda$.

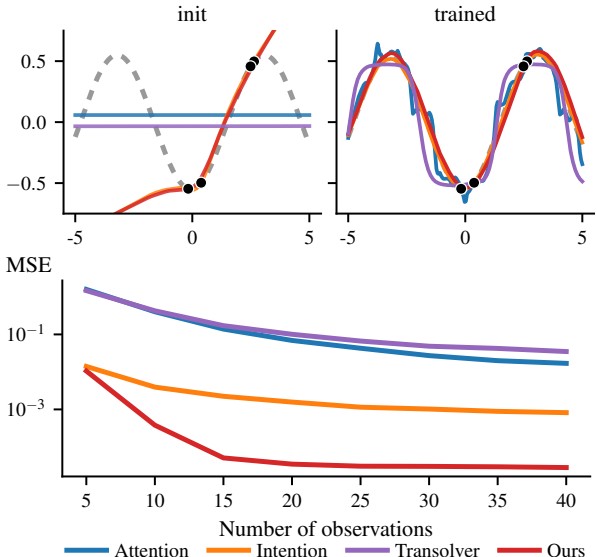

*Figure 2.* **Few-shot sinusoidal regression**. (Top) Predictions at initialization and after training on data with context length = 4 (black dots). Ground truth shown as a gray dotted line. $k$ in FUNCATTN and #slices in Transolver are set to 2. (Bottom) Generalization performance (MSE) across varying context sizes. Our method achieves the lowest MSE and scales most effectively with increasing context size.

**Proposition 4.5** (Local Lipschitz Continuity). *Let* $\mathbf{X} \in \mathbb{R}^{n \times d}$ *with* $\|\mathbf{X}\|_2 \leq B$, *and let* $\mathbf{Q} = \mathbf{X}\mathbf{W_Q}$, $\mathbf{K} = \mathbf{X}\mathbf{W_K}$, $\mathbf{V} = \mathbf{X}\mathbf{W_V}$. *For any* $\lambda > 0$, *the functional attention layer* $\mathcal{A}(\mathbf{X}) := \text{FUNCATTN}(\mathbf{Q}, \mathbf{K}, \mathbf{V})$ *satisfies*

$$\|\partial \mathcal{A}\|_F \leq \left( \frac{C_1}{\lambda} + \frac{C_2}{\lambda^2} \right) \|\Delta \mathbf{X}\|_F, \quad (11)$$

*where* $C_1, C_2 > 0$ *depend polynomially on* $B$, $n$, *and the weight norms* $\|\mathbf{W_Q}\|_2$, $\|\mathbf{W_K}\|_2$, $\|\mathbf{W_V}\|_2$, $\|\mathbf{W_\Phi}\|_2$, $\|\mathbf{W_\Psi}\|_2$, *and* $\|\partial \mathcal{A}\|_F$ *is the (Fréchet) differential of* $\mathcal{A}$.

A proof is provided in Appendix A.3. In particular, since the bound in (11) is linear in $\|\Delta \mathbf{X}\|_F$ with a finite prefactor for any $\lambda > 0$, local Lipschitz continuity of $\mathcal{A}$ follows immediately as a direct consequence of Proposition 4.5. We observe that the regularization parameter $\lambda$ controls our upper bound on the Lipschitz constant, formalizing the role of the Tikhonov term described in the Remark 4.2.

# 5. Experiments

In this section, we conduct extensive experiments to thoroughly validate FUNCATTN across diverse tasks, including few-shot regression, PDE solving, 3D point cloud segmentation and out-of-distribution generalization.

## 5.1. Sinusoidal Regression

Following (Finn et al., 2017), we consider few-shot sinusoidal regression where each task is a sine wave with random

amplitude $a \in [0.1, 5.0]$ and phase $\gamma \in [0, \pi]$. The goal is to predict function values at any query points given a set of fixed observations. We compare with scaled dot-product attention (Vaswani et al., 2017), Intention (Garnelo & Czarnecki, 2023), and Transolver (Wu et al., 2024) and choose a cross-attention architecture where keys, queries, and values are processed by separate encoders. To ensure fair comparison, we maintain similar parameter counts across all methods (cf. Appendix C.1 for details).

Fig. 2 (Top) illustrates the results before and after training (init vs. trained). Both scaled dot-product attention and Transolver initialize as a flat line, exhibiting no inductive bias for regression, while Intention and ours capture sinusoidal structure even before training. With only 4 observations, Ours regresses a smooth and accurate solution, whereas the scaled dot-product attention produces noisy predictions. Transolver estimates smoother solutions, however far from the target ground truth given as few as only four observations. Intention is the only baseline that achieves comparable performance in this low-data regime. Nevertheless, Ours consistently generalizes better across unseen numbers of observations, achieving errors that are up to three orders of magnitude lower than vanilla attention and Transolver, and one order of magnitude lower than Intention, as shown in Fig. 2 (Bottom). This result highlights the superior sample efficiency of FUNCATTN, which achieves lower error with only five observations than the scaled dot-product attention with forty observations. Indeed, while the scaled dot-product attention *interpolates* values through normalized affinities, FUNCATTN *regresses* spectral coefficients via least-squares. Consequently, interpolation-based methods rely on dense sampling of the input domain, whereas our approach leverages structural priors encoded in the learned basis, enabling higher accuracy and improved generalization from sparse observations.

## 5.2. PDE Solving

We evaluate on six PDE benchmarks from two physical domains: *Fluid mechanics* including subsurface flow (Darcy), turbulent flow (Navier-Stokes), and aerodynamics (Airfoil, Pipe); *Solid mechanics* including elastic (Elasticity) and plastic deformation (Plasticity). These tasks span point clouds, structured, and unstructured meshes (Li et al., 2021; 2023a).

We compare ours against strong neural operator methods, spanning frequency-domain approaches: e.g. FNO (Li et al., 2021), GEO-FNO (Li et al., 2023a), LSM (Wu et al., 2023), and attention-based architectures: e.g. Galerkin Transformer (Cao, 2021), which first applied attention to neural operator learning, and Transolver (Wu et al., 2024), which is the most recent physics-aware attention method. To ensure a fair comparison, we follow the experimental settings of

*Table 1.* **Quantitative results on PDE benchmarks.** Relative $L_2$ loss to ground truth ($\times 100$, $\downarrow$) is reported. Best results are in **bold**, second best are underlined. "/" indicates the method is not applicable. Ours reaches the SOTA results and outperforms in almost all datasets. See Tab. 10 in Appendix C.2 for implementation details of our FUNCATTN.

| | METHOD | ELASTICITY | AIRFOIL | DARCY | PIPE | NAVIER-STOKES | PLASTICITY |
|---|---|---|---|---|---|---|---|
| **FREQUENCY** | FNO (2021) | / | / | 1.08 | / | 15.56 | / |
| | WMT (2021A) | 3.59 | 0.75 | 0.82 | 0.77 | 15.41 | 0.76 |
| | U-FNO (2022) | 2.39 | 2.69 | 1.83 | 0.56 | 22.31 | 0.39 |
| | GEO-FNO (2023A) | 2.29 | 1.38 | 1.08 | 0.67 | 15.56 | 0.74 |
| | U-NO (2023) | 2.58 | 0.78 | 1.13 | 1.00 | 17.13 | 0.34 |
| | F-FNO (2023) | 2.63 | 0.78 | 0.77 | 0.70 | 23.22 | 0.47 |
| | LSM (2023) | 2.18 | 0.59 | 0.65 | 0.50 | 15.35 | 0.25 |
| **ATTENTION** | GALERKIN (2021) | 2.40 | 1.18 | 0.84 | 0.98 | 14.01 | 1.20 |
| | HT-NET (2022) | / | 0.65 | 0.79 | 0.54 | 18.47 | 3.33 |
| | OFORMER (2023C) | 1.83 | 1.83 | 1.24 | 1.68 | 17.05 | 0.17 |
| | GNOT (2023) | 0.86 | 0.76 | 1.05 | 0.47 | 13.80 | 3.36 |
| | FACTFORMER (2023D) | / | 0.71 | 1.09 | 0.60 | 12.14 | 3.12 |
| | ONO (2024) | 1.18 | 0.61 | 0.76 | 0.52 | 11.95 | 0.48 |
| | LNO (2024) | 0.73 | 0.54 | 0.60 | **0.25** | 8.45 | 0.31 |
| | TRANSOLVER (2024) | 0.64 | 0.53 | 0.57 | 0.31 | 9.44 | 0.13 |
| | **OURS** | **0.50** | **0.43** | **0.42** | 0.29 | **8.00** | **0.11** |

Transsolver (Wu et al., 2024). All experiments are conducted on a single Nvidia A40 GPU and repeated three times. See Appendix C.2 for details.

Tab. 1 shows that our approach achieves the best performance on five out of six PDE benchmarks, indicating strong and consistent performance across a diverse range of physical systems. Compared to Transsolver, a related transformer-based method (cf. Appendix A.4), our method yields relative improvements between $6\%$ and $26.3\%$. LNO performs competitively on the Pipe task; however, ours works on par in this task and consistently better in all remaining tasks. The good performance of LNO is likely due to its "physics-cross-attention", which is effective in capturing certain problem structures. These results suggest that efficiently learning an optimal linear operator between queries and keys provides a stronger inductive bias than softmax-based attention. Among other baselines, frequency-domain methods tend to struggle on complex geometries, where fixed spectral representations become less well aligned with the underlying domain structure. Earlier attention-based approaches, such as Galerkin Transformers, apply attention directly over mesh points, which can limit their ability to efficiently capture global, physics-relevant correlations. See Fig. 3 and Appendix E for visual examples.

**5.3. RNA Segmentation**

We also apply FUNCATTN to 3D tasks. To this end, we perform 3D segmentation tasks on the RNA dataset (Poulenard et al., 2019), which contains 640 ribosomal RNA structures from the Protein Data Bank (Berman et al., 2000). Each surface is represented as a point cloud of 4096 points, annotated with 259 functional categories. We apply random rotation augmentation for all methods taking raw coordi-

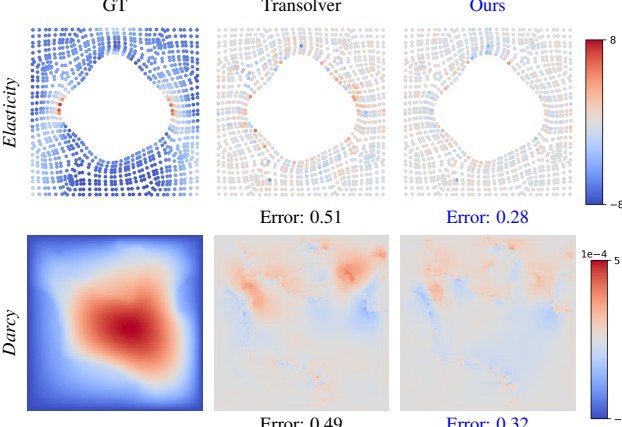

*Figure 3.* **PDE solving visualization.** Ground truth and error maps for Elasticity and Darcy benchmarks. Our method achieves lower error (relative $L_2$, $\times 100$) in both domains.

nates of point clouds as input. Tab. 2 summarizes the segmentation accuracy. FUNCATTN achieves the highest accuracy, outperforming both classical point cloud architectures, e.g. PointNet++ and recent operator-based approaches, e.g. DiffusionNet and Transsolver. We hypothesize that linear solving enables signed attention weights, which provides explicit contrastive capacity that is crucial for fine-grained segmentation.

**5.4. Out-of-Distribution (OOD) Generalization**

To evaluate the generalizability of learned representations beyond the training distribution, we conduct experiments on OOD airfoil design tasks (Bonnet et al., 2022). Unlike standard benchmarks where training and test samples share the same range of physical parameters, the OOD setting presents a more challenging scenario, in which the test set contains unseen Reynolds numbers and angles of attack. As

*Table 2.* **RNA point cloud segmentation.** "xyz" and "hks" indicates network input to be either xyz coordinates or heat kernel signatures (Sun et al., 2009). Ours achieves the best segmentation accuracy.

| Method | Accuracy ($\uparrow$) |
|---|---|
| PointNet++ (Qi et al., 2017) | 74.4% |
| PCNN (Atzmon et al., 2018) | 78.0% |
| SPHNet (Poulenard et al., 2019) | 80.1% |
| DiffusionNet - hks (Sharp et al., 2022) | 82.6% |
| DiffusionNet - xyz (Sharp et al., 2022) | 85.1% |
| Transolver - xyz (Wu et al., 2024) | 87.5% |
| **Ours** - xyz | **89.0%** |

*Table 3.* **OOD generalization on AirfRANS** (Bonnet et al., 2022). Relative error of the lift coefficient ($C_L$, %) and the Spearman's rank correlations ($\rho_L$, %) are reported as in Wu et al. (2024). All values are scaled by 100. Ours achieves the best generalization performance.

| MODELS | OOD REYNOLDS | | OOD ANGLES | |
|---|---|---|---|---|
| | $C_L$ ($\downarrow$) | $\rho_L$ ($\uparrow$) | $C_L$ ($\downarrow$) | $\rho_L$ ($\uparrow$) |
| SIMPLE MLP | 62.1 | 95.8 | 41.3 | 95.7 |
| GRAPHSAGE (2017) | 43.3 | 97.1 | 25.4 | 98.9 |
| POINTNET (2017) | 38.4 | 98.1 | 44.3 | 97.8 |
| GRAPH U-NET (2019) | 46.6 | 96.5 | 37.6 | 98.2 |
| MESHGRAPHNET (2020) | 177.2 | 76.3 | 65.3 | 89.3 |
| GNO (2023A) | 44.1 | 98.8 | 30.4 | 98.8 |
| GALERKIN (2021) | 46.2 | 98.3 | 38.1 | 98.2 |
| GNOT (2023) | 32.7 | 98.7 | 35.0 | 98.7 |
| GINO (2023B) | 41.8 | 96.5 | 25.8 | 99.2 |
| TRANSOLVER (2024) | 32.2 | 98.7 | 22.8 | 99.0 |
| **OURS** | **23.4** | **99.4** | **13.3** | **99.7** |

shown in Tab. 3, ours consistently generalizes better. On OOD Reynolds, ours achieves a relative error of 23.4% and Spearman's rank correlation of 99.4%, outperforming the closest competitor by a large margin of 8.8%. On OOD Angles, ours further reduces the relative error to 13.3% while maintaining the Spearman's rank correlation of 99.7%, improving upon the closest competitor by 9.5%. These results suggest that FUNCATTN not only fits the training data effectively, but also captures transferable physical patterns that generalize to unseen parameter regimes, highlighting the advantage of learning optimal linear map between functional spaces than tokenwise affinities.

### 5.5. Complex Geometry

Beyond the irregular meshes covered in Section 5.2, we test FUNCATTN on a challenging setting that combines a non-rectangular domain and a geometric re-entrant corner. Following (Tripura & Chakraborty, 2022), we evaluate on the 2D Darcy flow over a triangular domain with a notch, where the notch tip induces sharp local features in the solution field that are particularly challenging for fixed-basis spectral

*Table 4.* **2D Darcy flow with a triangular notch domain.** Relative $L^2$ error (%, $\downarrow$) is reported. Baseline results are taken from (Tripura & Chakraborty, 2022) (Table 3); † denotes our reproduction using the released code under a comparable parameter budget. Ours achieves the best performance on this singular-domain task.

| Method | Rel. $L^2$ ($\downarrow$) |
|---|---|
| DeepONet (Lu et al., 2021) | 2.64 |
| POD-DeepONet (Lu et al., 2022) | 1.00 |
| MWT (Gupta et al., 2021b) | 0.87 |
| dgFNO+ (Lu et al., 2022) | 7.82 |
| WNO$^\dagger$ (Tripura & Chakraborty, 2022) | 0.92 |
| **Ours** | **0.64** |

methods. As shown in Table 4, FUNCATTN achieves a relative $L^2$ error of 0.64%, a 30.9% relative improvement over WNO (Tripura & Chakraborty, 2022), which is specifically designed for complex-geometry PDEs. In contrast, grid-based spectral methods such as dgFNO+ perform substantially worse in this setting (7.82%), showing that fixed Cartesian bases are poorly suited to non-rectangular domains with sharp local features. These results indicate that the learned basis in FUNCATTN adapts to the underlying geometry and remains accurate near the notch tip.

*Table 5.* **Quantitative results of super-resolution task.** We utilize the 1D Burgers' equation dataset (Li et al., 2021). All modes are trained on 2048 grid points and tested on 8192. Relative $L_2$ error ($\times 1e3$) is reported. For a fair comparison, all methods are adjusted to have a similar number of parameters. Our method generalizes best to higher resolutions.

| Models | FNO | Galerkin | Transolver | **Ours** |
|---|---|---|---|---|
| Error ($\downarrow$) | 1.195 | 1.175 | 1.243 | **1.081** |

### 5.6. Zero-Shot Super-Resolution

A key property of neural operators is their discretization invariance, namely the ability to generalize across different mesh resolutions. We evaluate this by training on coarse grids and testing on finer resolutions. Specifically, we train on the 1D Burgers' equation dataset (Li et al., 2021) at a resolution of 2048 grid points, and evaluate on the full resolution of 8192 grid points without any fine-tuning. Tab. 5 shows that our FUNCATTN maintains strong performance under large resolution changes, demonstrating that the learned functional map captures resolution-independent structure of the underlying dynamic systems governed by PDE. For details, we refer the reader to Appendix C.2.

### 5.7. Ablation Study

**Number of Bases.** The number of bases $k$ controls the expressiveness of the learned functional attention. We study its effect on model performance on three PDE benchmarks. Tab. 6 shows the effect of varying the number of bases. We

*Table 6.* **Ablation on number of bases** $k$**.** We ablate on three PDE datasets and report the relative $L_2$ error ($\times 100, \downarrow$).

| Dataset | #Bases | | | | | |
|---|---|---|---|---|---|---|
| | 16 | 32 | 64 | 128 | 256 | 512 |
| Elasticity | 0.65 | 0.55 | 0.50 | 0.49 | **0.48** | 0.56 |
| Darcy | 0.49 | 0.45 | 0.42 | 0.44 | 0.43 | **0.41** |
| Airfoil | 0.51 | 0.52 | 0.43 | **0.42** | 0.47 | 0.48 |

*Table 7.* **Ablation on the choice of basis.** We study the effect of different bases on the performance of ours and two baselines, and report the relative $L_2$ ($\times 100, \downarrow$) on the Airfoil dataset. Note that the Fourier coefficients operate in the complex domain, where standard attention mechanisms are not directly applicable.

| Choice of Basis | Galerkin | Attention | Ours |
|---|---|---|---|
| Fourier | 0.65 | / | 0.51 |
| Learnable + orth. | 0.62 | 0.65 | 0.50 |
| **Learnable** | 0.59 | 0.53 | **0.43** |

observe that increasing the number of bases generally improves performance up to a certain point, after which performance slightly degrades due to potential overfitting. While larger values such as 256 or 512 yield slight improvements on specific datasets, they also introduce additional computational overhead. As practical guidance, $k = 64$ works as a robust default within 5% of the best across all benchmarks. For smoother fields (Darcy, Pipe), $k = 32$–$64$ suffices, for high-frequency fields (Elasticity, Navier-Stokes), $k = 128$–$256$ yields further gains. See Appendix C.2 for additional results.

**Choice of Basis.** We investigate the impact of different basis function by ablating on three choices: the fixed Fourier basis, the learnable basis as in Eq.(9) with additional orthogonal constraints, and the learnable basis as in Eq.(9). We evaluate these choices under three attention mechanisms: Galerkin attention (Cao, 2021), scaled dot-product attention (Vaswani et al., 2017), and FUNCATTN, and report the results in Tab. 7. Interestingly, freely learned basis Eq. (9) (last row in Tab.7) without enforcing orthogonality performs better, which coincides with observations in other works (Marin et al., 2020). This behavior may stem from the fact that optimizing over the orthogonal group is inherently more difficult than in Euclidean space, where commonly used gradient-based optimizers can more reliably identify good local minima. Moreover, even ours with fixed Fourier basis outperforms all baselines, underpinning the expressiveness of our functional attention framework. The preferable performance with freely learned basis is also observed in Galerkin and Attention.

## 6. Conclusion

By bridging functional map theory with attention mechanisms, we introduce a principled framework for capturing functional structure in operator learning. Rather than operating at the token level, our approach lifts attention to the functional space, enabling greater geometric flexibility and expressiveness. We further instantiate this functional correspondence framework through an optimization-based operator that links the query and key–value spaces, together with learnable bases. In addition, we provide a theoretical analysis of FUNCATTN, proving its Lipschitz continuity with respect to the input functions and thereby establishing its stability. Finally, we demonstrate the versatility of our method across a range of tasks. On PDE benchmarks, our model consistently achieves recent state-of-the-art methods in accuracy and exhibits superior generalization under domain shifts. In particular, our results on complex geometries further highlight the ability of FUNCATTN to adapt to nontrivial domains with sharp local features, underscoring its suitability for PDEs defined on general geometries. For 3D segmentation, we achieve higher accuracy than competing approaches. More broadly, this functional perspective opens new avenues for designing attention mechanisms that are structure-aware, resolution-invariant, and naturally suited to operator learning.

**Limitations & Future Works.** The learned basis uses a simple softmax projection; exploring more expressive or structured designs remains an open direction. While functional attention shows favorable inductive biases for operator learning, rigorous theoretical analysis, such as approximation guarantees or generalization bounds, is still needed. Formally connecting the compression ratio $k/n$ to approximation error would further strengthen our theoretical foundation. Additionally, other regularizations, e.g., $L_1$ penalties, may improve performance in specific applications. Finally, investigating functional attention in domains with less direct function-space interpretations, such as natural language processing, remains a promising future task.

## Impact Statement

This work advances operator learning by introducing a principled functional formulation of attention that improves robustness, efficiency, and generalization across resolutions and geometries. By enabling more reliable surrogate models for partial differential equations and geometric data, our approach may benefit scientific and engineering applications such as physical simulation, design optimization, and data-efficient modeling. We do not foresee significant negative societal impacts specific to this work beyond those common to data-driven modeling techniques. Also, training large data-driven models can be computationally expensive, with associated energy costs.

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

# – Appendix –

This appendix is organized as follows:

## A. Theoretical Insights

### A.1. Proof of Proposition 4.3

*Proof.* We prove each property separately.

**(i) Partition-of-Unity.** For any $x \in \Omega$ and $\tau > 0$, by definition of the softmax function:

$$\sum_{j=1}^{k} \phi_j^\tau(x) = \sum_{j=1}^{k} \frac{\exp(s_j(x)/\tau)}{\sum_{l=1}^{k} \exp(s_l(x)/\tau)} = \frac{\sum_{j=1}^{k} \exp(s_j(x)/\tau)}{\sum_{l=1}^{k} \exp(s_l(x)/\tau)} = 1 \tag{12}$$

Thus $\{\phi_j^\tau\}_{j=1}^{k}$ forms a partition of unity for all $\tau > 0$.

**(ii) Recovery of $P_0$ Elements.** Fix $x \in \Omega$ and assume without loss of generality that the scores have a unique maximum, i.e., there exists a unique $j^* = \arg\max_j s_j(x)$. Define the gap $\delta_l = s_{j^*}(x) - s_l(x) > 0$ for all $l \neq j^*$.

For the maximizing index $j^*$:

$$\phi_{j^*}^\tau(x) = \frac{\exp(s_{j^*}(x)/\tau)}{\sum_{l=1}^{k} \exp(s_l(x)/\tau)} = \frac{1}{1 + \sum_{l \neq j^*} \exp((s_l(x) - s_{j^*}(x))/\tau)} \tag{13}$$

$$= \frac{1}{1 + \sum_{l \neq j^*} \exp(-\delta_l/\tau)} \tag{14}$$

Since $\delta_l > 0$ for all $l \neq j^*$, we have $\exp(-\delta_l/\tau) \to 0$ as $\tau \to 0^+$. Therefore:

$$\lim_{\tau \to 0^+} \phi_{j^*}^\tau(x) = \frac{1}{1 + 0} = 1 \tag{15}$$

For any $j \neq j^*$:

$$\phi_j^\tau(x) = \frac{\exp(s_j(x)/\tau)}{\sum_{l=1}^{k} \exp(s_l(x)/\tau)} = \frac{\exp((s_j(x) - s_{j^*}(x))/\tau)}{1 + \sum_{l \neq j^*} \exp((s_l(x) - s_{j^*}(x))/\tau)} \tag{16}$$

$$= \frac{\exp(-\delta_j/\tau)}{1 + \sum_{l \neq j^*} \exp(-\delta_l/\tau)} \tag{17}$$

Since the numerator $\exp(-\delta_j/\tau) \to 0$ and the denominator $\to 1$ as $\tau \to 0^+$:

$$\lim_{\tau \to 0^+} \phi_j^\tau(x) = 0 \tag{18}$$

Combining both cases, we obtain:

$$\lim_{\tau \to 0^+} \phi_j^\tau(x) = \begin{cases} 1 & \text{if } j = \arg\max_l s_l(x) \\ 0 & \text{otherwise} \end{cases} = \mathbf{1}_{\Lambda_j}(x) \tag{19}$$

where $\Lambda_j = \{x \in \Omega : s_j(x) > s_l(x), \forall l \neq j\}$. This recovers the classical $P_0$ piecewise constant basis with hard partitioning. $\qquad\square$

## A.2. Approximated Integral Neural Operator

In this section, we establish that our method approximates an integral neural operator. The proof follows a similar technique to that of Transolver (Wu et al., 2024)

**Lemma A.1** (Wu et al. (2024)). *Suppose that $\Omega$ is a countable domain, the reduced domain $\Omega_{\mathrm{spec}}$ is isomorphic to $\Omega$.*

**Lemma A.2.** *The operator $\left[\mathbf{QK}^\top (\mathbf{KK}^\top + \lambda \mathbf{I}_n)^{-1}\right] \mathbf{V}$ can be interpreted as a Monte-Carlo discretization of a regularized integral operator.*

*Proof.* Given input function $\boldsymbol{u} : \Omega \to \mathbb{R}^C$, define the key Gram kernel $h(\xi, \xi') := \boldsymbol{k}(\xi)^\top \boldsymbol{k}(\xi')$ where $\boldsymbol{k}(\xi) = \mathbf{W}_k \boldsymbol{u}(\xi)$, and the associated integral operator

$$(\mathcal{H}f)(\xi) := \int_\Omega h(\xi, \xi') f(\xi') \,\mathrm{d}\xi'. \tag{20}$$

The regularized integral operator $\mathcal{G}_\lambda$ on the function space $\Omega \to \mathbb{R}^C$ is defined as:

$$\mathcal{G}_\lambda(\boldsymbol{u})(\mathbf{g}^*) = \int_\Omega \kappa_\lambda(\mathbf{g}^*, \xi) \boldsymbol{v}(\xi) \,\mathrm{d}\xi, \tag{21}$$

where $\boldsymbol{q}(\xi) = \mathbf{W}_q \boldsymbol{u}(\xi)$, $\boldsymbol{v}(\xi) = \mathbf{W}_v \boldsymbol{u}(\xi)$, and the regularized kernel is

$$\kappa_\lambda(\mathbf{g}^*, \xi) := \boldsymbol{q}(\mathbf{g}^*)^\top \boldsymbol{k}(\xi) \cdot [(\mathcal{H} + \lambda I)^{-1}](\xi). \tag{22}$$

Suppose there are $n$ discretized mesh points $\{\mathbf{g}_1, \cdots, \mathbf{g}_n\}$ with $\mathbf{g}_i \in \Omega$. Approximating $\mathcal{H}$ by Monte-Carlo gives

$$(\mathcal{H}f)(\mathbf{g}_i) \approx \frac{|\Omega|}{n} \sum_{j=1}^n \boldsymbol{k}(\mathbf{g}_i)^\top \boldsymbol{k}(\mathbf{g}_j) \, f(\mathbf{g}_j) \quad \rightsquigarrow \quad \mathcal{H} \approx \frac{|\Omega|}{n} \mathbf{KK}^\top, \tag{23}$$

where $\mathbf{K} \in \mathbb{R}^{n \times C}$ stacks $\boldsymbol{k}(\mathbf{g}_i)^\top$ row-wise. Applying the same approximation to the outer integral and absorbing the constant $\frac{|\Omega|}{n}$ into $\lambda$, we obtain

$$\mathcal{G}_\lambda(\boldsymbol{u})(\mathbf{g}^*) \approx \left[\mathbf{QK}^\top (\mathbf{KK}^\top + \lambda \mathbf{I}_n)^{-1}\right] \mathbf{V}, \tag{24}$$

which completes the proof. $\qquad\square$

**Theorem A.3** (**Functional Attention is equivalent to learnable integral on $\Omega$**). *Given input function $\boldsymbol{u} : \Omega \to \mathbb{R}^C$, Functional Attention approximates an integral operator $\mathcal{G}$ on $\Omega$:*

$$\mathcal{G}(\boldsymbol{u})(\mathbf{g}^*) = \int_\Omega \kappa(\mathbf{g}^*, \boldsymbol{\xi}) \boldsymbol{v}(\boldsymbol{\xi}) \mathrm{d}\boldsymbol{\xi} \tag{25}$$

*where $\kappa(\cdot, \cdot)$ is a learnable kernel on $\Omega \times \Omega$.*

*Proof.* Following a similar argument as in Wu et al. (2024), by Lemma A.1 and Lemma A.2, Functional Attention $\textsc{FuncAttn}(\mathbf{Q}, \mathbf{K}, \mathbf{V}) = \boldsymbol{\Phi}\,\mathbf{C}\,\boldsymbol{\Psi}^\top \mathbf{V}$ corresponds to a Monte-Carlo discretization of the integral operator (25) with kernel $\kappa(\mathbf{g}_i, \mathbf{g}_j) = (\boldsymbol{\Phi}\,\mathbf{C}\,\boldsymbol{\Psi}^\top)_{ij}$. $\qquad\square$

## A.3. Proof of Proposition 4.5

*Proof.* We compute the (Fréchet) differential $\partial\mathcal{A}$ of $\mathcal{A}(\mathbf{X}) = \textsc{FuncAttn}(\mathbf{Q}, \mathbf{K}, \mathbf{V})$ in the direction $\Delta\mathbf{X}$ and bound its Frobenius norm. Throughout, $\|\cdot\|_2$ denotes the spectral norm and we repeatedly use the submultiplicative inequality $\|\mathbf{AB}\|_F \leq \|\mathbf{A}\|_2 \|\mathbf{B}\|_F$.

**Step 1: Preliminary norm estimates.** By construction (Eq. 9), each row of $\boldsymbol{\Phi}(\mathbf{X}), \boldsymbol{\Psi}(\mathbf{X}) \in \mathbb{R}^{n \times k}$ is the output of a softmax along the $k$ dimension, hence has $\ell_2$ norm at most 1. Therefore

$$\|\boldsymbol{\Phi}\|_2 \leq \|\boldsymbol{\Phi}\|_F \leq \sqrt{n}, \qquad \|\boldsymbol{\Psi}\|_2 \leq \|\boldsymbol{\Psi}\|_F \leq \sqrt{n}. \tag{26}$$

Combined with $\|\mathbf{X}\|_2 \leq B$,

$$\|\mathbf{Q}\|_2 \leq B\|\mathbf{W_Q}\|_2, \qquad \|\mathbf{K}\|_2 \leq B\|\mathbf{W_K}\|_2, \qquad \|\mathbf{V}\|_2 \leq B\|\mathbf{W_V}\|_2, \qquad (27)$$

$$\|\widetilde{\mathbf{Q}}\|_2 \leq \sqrt{n}\, B\, \|\mathbf{W_Q}\|_2, \qquad \|\widetilde{\mathbf{K}}\|_2 \leq \sqrt{n}\, B\, \|\mathbf{W_K}\|_2, \qquad \|\widetilde{\mathbf{V}}\|_2 \leq \sqrt{n}\, B\, \|\mathbf{W_V}\|_2. \qquad (28)$$

Let $\widetilde{\mathbf{S}} := \widetilde{\mathbf{K}}\widetilde{\mathbf{K}}^\top + \lambda \mathbf{I}_k$. Since $\widetilde{\mathbf{K}}\widetilde{\mathbf{K}}^\top \succeq 0$ and $\lambda > 0$,

$$\widetilde{\mathbf{S}} \succeq \lambda\, \mathbf{I}_k \quad \implies \quad \|\widetilde{\mathbf{S}}^{-1}\|_2 \leq \tfrac{1}{\lambda}. \qquad (29)$$

**Step 2: Lipschitz constants of the building blocks.** From $\mathbf{Q} = \mathbf{X}\mathbf{W_Q}$ (and analogously for $\mathbf{K}, \mathbf{V}$),

$$\|\partial\mathbf{Q}\|_F \leq \|\mathbf{W_Q}\|_2\|\Delta\mathbf{X}\|_F, \quad \|\partial\mathbf{K}\|_F \leq \|\mathbf{W_K}\|_2\|\Delta\mathbf{X}\|_F, \quad \|\partial\mathbf{V}\|_F \leq \|\mathbf{W_V}\|_2\|\Delta\mathbf{X}\|_F. \qquad (30)$$

For the row-wise softmax basis, a standard result (Gao & Pavel, 2017, Prop. 4) provides $L_\mathbf{\Phi}, L_\mathbf{\Psi} > 0$ (depends only on its temperature), composing with the linear pre-activation gives such that

$$\|\partial\mathbf{\Phi}\|_F \leq L_\mathbf{\Phi}\, \|\mathbf{W_\Phi}\|_2\, \|\Delta\mathbf{X}\|_F, \qquad \|\partial\mathbf{\Psi}\|_F \leq L_\mathbf{\Psi}\, \|\mathbf{W_\Psi}\|_2\, \|\Delta\mathbf{X}\|_F. \qquad (31)$$

Applying the product rule to $\widetilde{\mathbf{Q}} = \mathbf{\Phi}^\top \mathbf{Q}$,

$$\begin{aligned}
\|\partial\widetilde{\mathbf{Q}}\|_F &\leq \|\partial\mathbf{\Phi}\|_F\, \|\mathbf{Q}\|_2 + \|\mathbf{\Phi}\|_2\, \|\partial\mathbf{Q}\|_F \\
&\leq \left(L_\mathbf{\Phi}\, B\, \|\mathbf{W_\Phi}\|_2 + \sqrt{n}\right)\|\mathbf{W_Q}\|_2\, \|\Delta\mathbf{X}\|_F \ =: \ \alpha_\mathbf{\Phi}\, \|\mathbf{W_Q}\|_2\, \|\Delta\mathbf{X}\|_F,
\end{aligned} \qquad (32)$$

and similarly, since $\widetilde{\mathbf{K}} = \mathbf{\Psi}^\top\mathbf{K}$ and $\widetilde{\mathbf{V}} = \mathbf{\Psi}^\top\mathbf{V}$,

$$\|\partial\widetilde{\mathbf{K}}\|_F \leq \alpha_\mathbf{\Psi}\, \|\mathbf{W_K}\|_2\, \|\Delta\mathbf{X}\|_F, \qquad \|\partial\widetilde{\mathbf{V}}\|_F \leq \alpha_\mathbf{\Psi}\, \|\mathbf{W_V}\|_2\, \|\Delta\mathbf{X}\|_F, \qquad (33)$$

where we set

$$\alpha_\mathbf{\Phi} := L_\mathbf{\Phi}\, B\, \|\mathbf{W_\Phi}\|_2 + \sqrt{n}, \qquad \alpha_\mathbf{\Psi} := L_\mathbf{\Psi}\, B\, \|\mathbf{W_\Psi}\|_2 + \sqrt{n}. \qquad (34)$$

**Step 3: Differential of $\mathcal{A}$.** Write $\mathcal{A} = \mathbf{\Phi} \cdot \mathcal{B}$ with $\mathcal{B} := \widetilde{\mathbf{Q}}\widetilde{\mathbf{K}}^\top\widetilde{\mathbf{S}}^{-1}\widetilde{\mathbf{V}}$. Then

$$\partial\mathcal{A} = \underbrace{\partial\mathbf{\Phi} \cdot \mathcal{B}}_{T_1} + \mathbf{\Phi} \cdot \partial\mathcal{B}, \qquad (35)$$

and the product rule yields

$$\begin{aligned}
\partial\mathcal{B} = &\underbrace{\partial\widetilde{\mathbf{Q}}\,\widetilde{\mathbf{K}}^\top\widetilde{\mathbf{S}}^{-1}\widetilde{\mathbf{V}}}_{T_2} + \underbrace{\widetilde{\mathbf{Q}}\,(\partial\widetilde{\mathbf{K}})^\top\widetilde{\mathbf{S}}^{-1}\widetilde{\mathbf{V}}}_{T_3} \\
&+ \underbrace{\widetilde{\mathbf{Q}}\,\widetilde{\mathbf{K}}^\top\,(\partial\widetilde{\mathbf{S}}^{-1})\,\widetilde{\mathbf{V}}}_{T_4} + \underbrace{\widetilde{\mathbf{Q}}\,\widetilde{\mathbf{K}}^\top\widetilde{\mathbf{S}}^{-1}\,\partial\widetilde{\mathbf{V}}}_{T_5},
\end{aligned} \qquad (36)$$

where, using $\partial(\mathbf{A}^{-1}) = -\mathbf{A}^{-1}(\partial\mathbf{A})\mathbf{A}^{-1}$,

$$\partial\widetilde{\mathbf{S}}^{-1} = -\widetilde{\mathbf{S}}^{-1}\left[\partial\widetilde{\mathbf{K}}\,\widetilde{\mathbf{K}}^\top + \widetilde{\mathbf{K}}\,(\partial\widetilde{\mathbf{K}})^\top\right]\widetilde{\mathbf{S}}^{-1}. \qquad (37)$$

**Step 4: Term-by-term bounds.** Set $\Theta := \|\mathbf{W_Q}\|_2\|\mathbf{W_K}\|_2\|\mathbf{W_V}\|_2$. We bound each term in turn.

*(i) Bound for $T_1$.* Using $\|T_1\|_F \leq \|\partial\mathbf{\Phi}\|_F\|\mathcal{B}\|_2$ and $\|\mathcal{B}\|_2 \leq \|\widetilde{\mathbf{Q}}\|_2\|\widetilde{\mathbf{K}}\|_2\|\widetilde{\mathbf{S}}^{-1}\|_2\|\widetilde{\mathbf{V}}\|_2 \leq n^{3/2}B^3\Theta/\lambda$,

$$\|T_1\|_F \leq \frac{L_\mathbf{\Phi}\, n^{3/2}B^3\, \|\mathbf{W_\Phi}\|_2\, \Theta}{\lambda}\, \|\Delta\mathbf{X}\|_F. \qquad (38)$$

*(ii) Bound for $\mathbf{\Phi} \cdot T_2$.*

$$\begin{aligned}
\|\mathbf{\Phi}T_2\|_F &\leq \|\mathbf{\Phi}\|_2 \|\partial\widetilde{\mathbf{Q}}\|_F \|\widetilde{\mathbf{K}}\|_2 \|\widetilde{\mathbf{S}}^{-1}\|_2 \|\widetilde{\mathbf{V}}\|_2 \\
&\leq \sqrt{n} \cdot \alpha_{\mathbf{\Phi}}\|\mathbf{W}_{\mathbf{Q}}\|_2 \cdot \sqrt{n}\,B\,\|\mathbf{W}_{\mathbf{K}}\|_2 \cdot \frac{1}{\lambda} \cdot \sqrt{n}\,B\,\|\mathbf{W}_{\mathbf{V}}\|_2 \|\Delta\mathbf{X}\|_F \\
&= \frac{n^{3/2}B^2\alpha_{\mathbf{\Phi}}\Theta}{\lambda}\|\Delta\mathbf{X}\|_F.
\end{aligned} \tag{39}$$

*(iii) Bound for $\mathbf{\Phi} \cdot T_3$.* Analogously,

$$\|\mathbf{\Phi}T_3\|_F \leq \frac{n^{3/2}B^2\alpha_{\mathbf{\Psi}}\Theta}{\lambda}\|\Delta\mathbf{X}\|_F. \tag{40}$$

*(iv) Bound for $\mathbf{\Phi} \cdot T_5$.*

$$\|\mathbf{\Phi}T_5\|_F \leq \frac{n^{3/2}B^2\alpha_{\mathbf{\Psi}}\Theta}{\lambda}\|\Delta\mathbf{X}\|_F. \tag{41}$$

*(v) Bound for $\mathbf{\Phi} \cdot T_4$.* Expanding $\partial\widetilde{\mathbf{S}}^{-1}$ and using the triangle inequality,

$$\begin{aligned}
\|\mathbf{\Phi}T_4\|_F &\leq 2\,\|\mathbf{\Phi}\|_2 \|\widetilde{\mathbf{Q}}\|_2 \|\widetilde{\mathbf{K}}\|_2 \|\widetilde{\mathbf{S}}^{-1}\|_2 \|\partial\widetilde{\mathbf{K}}\|_F \|\widetilde{\mathbf{K}}\|_2 \|\widetilde{\mathbf{S}}^{-1}\|_2 \|\widetilde{\mathbf{V}}\|_2 \\
&\leq 2\sqrt{n} \cdot \sqrt{n}B\|\mathbf{W}_{\mathbf{Q}}\|_2 \cdot \sqrt{n}B\|\mathbf{W}_{\mathbf{K}}\|_2 \cdot \frac{1}{\lambda} \cdot \alpha_{\mathbf{\Psi}}\|\mathbf{W}_{\mathbf{K}}\|_2 \cdot \\
&\qquad \cdot \sqrt{n}B\|\mathbf{W}_{\mathbf{K}}\|_2 \cdot \frac{1}{\lambda} \cdot \sqrt{n}B\|\mathbf{W}_{\mathbf{V}}\|_2 \|\Delta\mathbf{X}\|_F \\
&= \frac{2\,n^3 B^4\alpha_{\mathbf{\Psi}}\,\|\mathbf{W}_{\mathbf{Q}}\|_2\|\mathbf{W}_{\mathbf{K}}\|_2^3\|\mathbf{W}_{\mathbf{V}}\|_2}{\lambda^2}\|\Delta\mathbf{X}\|_F.
\end{aligned} \tag{42}$$

**Step 5: Combining the bounds** Summing $T_1$, $\mathbf{\Phi}T_2$, $\mathbf{\Phi}T_3$, $\mathbf{\Phi}T_5$ (all $\mathcal{O}(1/\lambda)$) and $\mathbf{\Phi}T_4$ ($\mathcal{O}(1/\lambda^2)$),

$$\|\partial\mathcal{A}\|_F \leq \left(\frac{C_1}{\lambda} + \frac{C_2}{\lambda^2}\right)\|\Delta\mathbf{X}\|_F, \tag{43}$$

with the explicit constants

$$\begin{aligned}
C_1 &= n^{3/2}B^2\Theta\left(L_{\mathbf{\Phi}}\,B\,\|\mathbf{W}_{\mathbf{\Phi}}\|_2 + \alpha_{\mathbf{\Phi}} + 2\,\alpha_{\mathbf{\Psi}}\right) \\
&= n^{3/2}B^2\Theta\left(2L_{\mathbf{\Phi}}\,B\,\|\mathbf{W}_{\mathbf{\Phi}}\|_2 + 2L_{\mathbf{\Psi}}\,B\,\|\mathbf{W}_{\mathbf{\Psi}}\|_2 + 3\sqrt{n}\right),
\end{aligned} \tag{44}$$

$$\begin{aligned}
C_2 &= 2\,n^3 B^4\,\alpha_{\mathbf{\Psi}}\,\|\mathbf{W}_{\mathbf{Q}}\|_2\|\mathbf{W}_{\mathbf{K}}\|_2^3\|\mathbf{W}_{\mathbf{V}}\|_2 \\
&= 2\,n^3 B^4\left(L_{\mathbf{\Psi}}\,B\,\|\mathbf{W}_{\mathbf{\Psi}}\|_2 + \sqrt{n}\right)\|\mathbf{W}_{\mathbf{Q}}\|_2\|\mathbf{W}_{\mathbf{K}}\|_2^3\|\mathbf{W}_{\mathbf{V}}\|_2,
\end{aligned} \tag{45}$$

where $\Theta = \|\mathbf{W}_{\mathbf{Q}}\|_2\|\mathbf{W}_{\mathbf{K}}\|_2\|\mathbf{W}_{\mathbf{V}}\|_2$. Both $C_1, C_2 > 0$ and depend polynomially on $B$, $n$, $\|\mathbf{W}_{\mathbf{Q}}\|_2$, $\|\mathbf{W}_{\mathbf{K}}\|_2$, $\|\mathbf{W}_{\mathbf{V}}\|_2$, $\|\mathbf{W}_{\mathbf{\Phi}}\|_2$, $\|\mathbf{W}_{\mathbf{\Psi}}\|_2$ (with multiplicative softmax-Lipschitz factors $L_{\mathbf{\Phi}}, L_{\mathbf{\Psi}}$). This proves (11). $\qquad\square$

### A.4. Transolver versus FUNCATTN.

At first glance, Transolver and FUNCATTN share a similar high-level structure: both models project the input onto a set of learned basis functions, perform interactions in a reduced coefficient space, and reconstruct the output via an inverse projection (*deslicing* step, in Transolver). Despite this apparent similarity, the two approaches differ fundamentally in their modeling perspective, as shown in Fig. 4. Transolver learns physics-aware bases that are explicitly tied to the discretized domain and are used to construct physically meaningful tokens $\mathbf{Q}, \mathbf{K}, \mathbf{V}$, on which standard attention is applied. In contrast, FUNCATTN operates at a more abstract functional level: attention is formulated directly as a mapping between function spaces, without relying on physics-specific tokenization or domain-dependent slicing. This functional abstraction decouples the attention mechanism from the underlying discretization and enables a more general and flexible operator representation, as demonstrated in Section 5.

### A.5. Connection with IntentionNet

In this section, we show that Intention (Garnelo & Czarnecki, 2023) can be recovered as a special case of Functional Attention under a restrictive choice of basis.

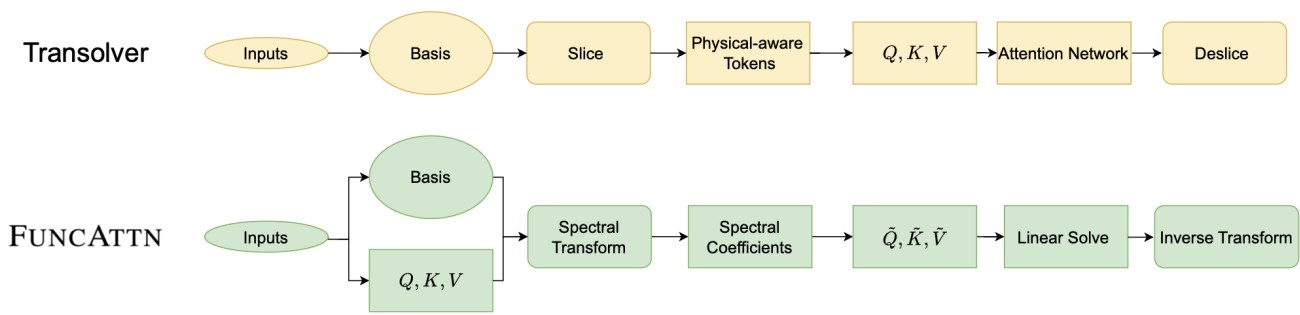

*Figure 4.* **Overall design of Transolver (Wu et al., 2024) and FUNCATTN.**

**Background: Intention** Intention (Garnelo & Czarnecki, 2023) was proposed as an attention mechanism capable of representing regularized least squares fitting. Given queries $\mathbf{Q} \in \mathbb{R}^{n \times d}$, keys $\mathbf{K} \in \mathbb{R}^{n \times d}$, and values $\mathbf{V} \in \mathbb{R}^{n \times d}$, Intention computes:

$$\text{Intention}(\mathbf{Q}, \mathbf{K}, \mathbf{V}) = \mathbf{Q}(\mathbf{K}^\top \mathbf{K} + \lambda \mathbf{I}_d)^{-1} \mathbf{K}^\top \mathbf{V} \tag{46}$$

**Functional Attention Recovers Intention** We now show that Intention is a special case of Functional Attention when we choose any *orthonormal basis* spanning the full space.

**Proposition A.4** (Intention as a Special Case). *Let $\mathbf{\Phi} = \mathbf{\Psi} \in \mathbb{R}^{n \times n}$ be any orthonormal basis, i.e., $\mathbf{\Phi}^\top \mathbf{\Phi} = \mathbf{\Phi} \mathbf{\Phi}^\top = \mathbf{I}_n$. Then Functional Attention reduces to Intention:*

$$\text{FUNCATTN}(\mathbf{Q}, \mathbf{K}, \mathbf{V}) = \mathbf{Q}(\mathbf{K}^\top \mathbf{K} + \lambda \mathbf{I}_d)^{-1} \mathbf{K}^\top \mathbf{V} = \text{Intention}(\mathbf{Q}, \mathbf{K}, \mathbf{V}) \tag{47}$$

*Proof.* With orthonormal $\mathbf{\Phi} = \mathbf{\Psi}$ satisfying $\mathbf{\Phi}^\top \mathbf{\Phi} = \mathbf{\Phi} \mathbf{\Phi}^\top = \mathbf{I}_n$, the spectral coefficients are:

$$\tilde{\mathbf{Q}} = \mathbf{\Phi}^\top \mathbf{Q}, \quad \tilde{\mathbf{K}} = \mathbf{\Phi}^\top \mathbf{K}, \quad \tilde{\mathbf{V}} = \mathbf{\Phi}^\top \mathbf{V} \tag{48}$$

Substituting into the Functional Attention formula (8):

$$\text{FUNCATTN}(\mathbf{Q}, \mathbf{K}, \mathbf{V}) = \mathbf{\Phi} \left[ \tilde{\mathbf{Q}} \tilde{\mathbf{K}}^\top (\tilde{\mathbf{K}} \tilde{\mathbf{K}}^\top + \lambda \mathbf{I}_n)^{-1} \right] \tilde{\mathbf{V}} \tag{49}$$

$$= \mathbf{\Phi} \left[ \mathbf{\Phi}^\top \mathbf{Q} \mathbf{K}^\top \mathbf{\Phi} (\mathbf{\Phi}^\top \mathbf{K} \mathbf{K}^\top \mathbf{\Phi} + \lambda \mathbf{I}_n)^{-1} \right] \mathbf{\Phi}^\top \mathbf{V} \tag{50}$$

Since $\mathbf{\Phi}$ is orthonormal, we have $\mathbf{\Phi}^\top \mathbf{K} \mathbf{K}^\top \mathbf{\Phi} + \lambda \mathbf{I}_n = \mathbf{\Phi}^\top (\mathbf{K} \mathbf{K}^\top + \lambda \mathbf{I}_n) \mathbf{\Phi}$, and its inverse is $\mathbf{\Phi}^\top (\mathbf{K} \mathbf{K}^\top + \lambda \mathbf{I}_n)^{-1} \mathbf{\Phi}$. Thus:

$$\text{FUNCATTN}(\mathbf{Q}, \mathbf{K}, \mathbf{V}) = \mathbf{\Phi} \mathbf{\Phi}^\top \mathbf{Q} \mathbf{K}^\top \mathbf{\Phi} \mathbf{\Phi}^\top (\mathbf{K} \mathbf{K}^\top + \lambda \mathbf{I}_n)^{-1} \mathbf{\Phi} \mathbf{\Phi}^\top \mathbf{V} \tag{51}$$

$$= \mathbf{Q} \mathbf{K}^\top (\mathbf{K} \mathbf{K}^\top + \lambda \mathbf{I}_n)^{-1} \mathbf{V} \tag{52}$$

where we used $\mathbf{\Phi} \mathbf{\Phi}^\top = \mathbf{I}_n$ three times. Finally, applying the Woodbury identity:

$$\mathbf{K}^\top (\mathbf{K} \mathbf{K}^\top + \lambda \mathbf{I}_n)^{-1} = (\mathbf{K}^\top \mathbf{K} + \lambda \mathbf{I}_d)^{-1} \mathbf{K}^\top \tag{53}$$

we obtain:

$$\text{FUNCATTN}(\mathbf{Q}, \mathbf{K}, \mathbf{V}) = \mathbf{Q}(\mathbf{K}^\top \mathbf{K} + \lambda \mathbf{I}_d)^{-1} \mathbf{K}^\top \mathbf{V} = \text{Intention}(\mathbf{Q}, \mathbf{K}, \mathbf{V}) \tag{54}$$

$\square$

## B. Complexity Analysis

### B.1. Theoretical Complexity

The steps in computing FUNCATTN consist of matrix multiplications and a (small) matrix inversion when solving the linear system for functional transport. Below, we break down each step and analyze its computational complexity.

**Basis Computation:** Our learned basis is adaptive to the input and has to be recomputed whenever the input changes. It has a complexity of $O(ndk)$ for the linear transformation and $O(nk)$ for the softmax operation.

**Latent Projection:** $\mathbf{Q}$, $\mathbf{K}$, $\mathbf{V}$ are projected to the corresponding latent spaces with bases $\mathbf{\Phi}$ and $\mathbf{\Psi}$, which are related by a linear transport $\mathbf{C}$. The three projections $\tilde{\mathbf{Q}} = \mathbf{\Phi}^T\mathbf{Q}$, $\tilde{\mathbf{K}} = \mathbf{\Psi}^T\mathbf{K}$, $\tilde{\mathbf{V}} = \mathbf{\Psi}^T\mathbf{V}$ have linear complexity $O(ndk)$.

**Linear Solve:** the functional transport $\mathbf{C}^*$ is computed by Eq. (7), which only involves operation on small matrices, since $k, d \ll n$. Thanks to the Woodybury matrix identity (Harville, 1997), Eq. (7) can be reformulated as $\mathbf{C}^* = \tilde{\mathbf{Q}}\tilde{\mathbf{K}}^\top(\tilde{\mathbf{K}}\tilde{\mathbf{K}}^\top + \lambda\mathbf{I}_k)^{-1} = \tilde{\mathbf{Q}}(\tilde{\mathbf{K}}^\top\tilde{\mathbf{K}} + \lambda\mathbf{I}_d)^{-1}\tilde{\mathbf{K}}^\top$. This leads to the fact that we only have to invert the smaller matrix, either $d \times d$ or $k \times k$, and obtain numerically identical results. This has a direct position impact computationally.

The computation of $\tilde{\mathbf{K}}\tilde{\mathbf{K}}^T + \lambda\mathbf{I}_k$ has complexity $O(dk^2)$, followed by the inversion with complexity $O(k^3)$ if no additional structure can be exploited. Additionally, the computation of $\tilde{\mathbf{Q}}\tilde{\mathbf{K}}^T$ has complexity $O(dk^2)$ and the final matrix multiplication has complexity $O(k^3)$. This results in a complexity of $O(dk^2 + k^3)$, which is independent of the possibly large context length $n$.

Alternatively, one can compute $\tilde{\mathbf{K}}^\top\tilde{\mathbf{K}} + \lambda\mathbf{I}_d$ with complexity $O(d^2k)$, followed by the inversion with complexity $O(d^3)$. This results in a complexity of $O(d^2k + d^3)$, which is preferable when $d < k$.

**Transport and Back-Projection:** The optimal transport $\mathbf{C}^*$ is used to transport $\tilde{\mathbf{V}}$ to the query space by matrix multiplication and has a complexity $O(dk^2)$, after which it is multiplied with the learned basis for the query space $\mathbf{\Phi}$ and has a complexity $O(ndk)$.

In summary, the computation complexity of FUNCATTN is $O(ndk + dk\min(k, d) + \min(k, d)^3)$, which is linear in $n$ and $d$, and cubic in $k$, which is typically small in practice and is set to $64$ and proven to be effective in our case. In contrast to the classic scaled dot-product attention, which has a cubic complexity in $n$, FUNCATTN is much more efficient, both in terms of runtime and number of tokens.

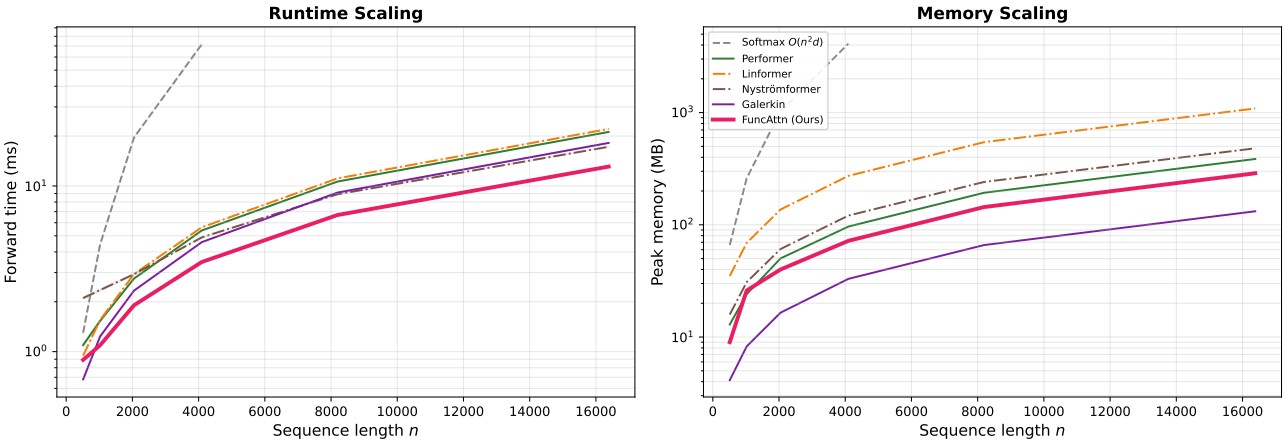

*Figure 5.* **Runtime and memory scaling.** Forward-pass time (left) and peak GPU memory (right) plots of sequence length $n$, with $d = 128$, $k = 64$. Softmax attention grows quadratically, whereas FUNCATTN exhibits the predicted linear scaling and outperforms other linear-attention baselines at large $n$.

### B.2. Empirical Runtime and Memory Scaling

To complement the theoretical analysis, we benchmark the forward-pass runtime and peak GPU memory of FUN-CATTN against representative baselines: the standard softmax attention (Vaswani et al., 2017), Performer (Choromanski et al., 2020), Linformer (Wang et al., 2020), Nyströmformer (Xiong et al., 2021), and Galerkin attention (Cao, 2021). We sweep the sequence length $n \in \{2^7, 2^8, \ldots, 2^{14}\}$ with fixed feature dimension $d = 128$, basis count $k = 64$, batch size 1, and a single forward pass on an NVIDIA A40 GPU; measurements include the adaptive basis computation. As shown in Fig. 5, softmax attention exhibits the expected quadratic growth in both runtime and memory, becoming prohibitively expensive at long contexts. In contrast, FUNCATTN scales linearly in $n$, matching our theoretical analysis. While the other linear-attention

variants share the same asymptotic $O(n)$ trend, FUNCATTN consistently achieves the smallest wall-clock time and memory footprint at large $n$, owing to its compact $k \times k$ operator and the absence of per-token softmax normalization. The gap widens with $n$, making FUNCATTN particularly suited to high-resolution operator learning.

## C. Experiment Details

### C.1. Regression Task

**Task.** We consider a meta-learning setting from (Finn et al., 2017) where each task corresponds to a sinusoidal function $f(x) = \alpha \sin(x - \gamma)$ defined on $x \in [-6, 6]$. The amplitude $\alpha$ and phase $\gamma$ are sampled uniformly from $[0.1, 5]$ and $[0, \pi]$, respectively. For each task, we observe a support set of $K$ randomly sampled input-output pairs. The goal is to learn a predictor that generalizes to arbitrary query locations given only the support set. Performance is measured by mean squared error on unseen query points, averaged over tasks.

**Model Framework.** All methods share the same encoder-decoder architecture and differ only in the attention mechanism $\mathcal{A}$:

$$\hat{y} = f_{\text{dec}}\big(\mathcal{A}(f_{\text{enc}}(\text{support}), \text{query})\big). \tag{55}$$

Table 8 details each configuration.

**Training Details.** For Attention and Intention, we adopt the hyperparameters from (Garnelo & Czarnecki, 2023). To ensure fair comparison, we maintain a similar number of parameters across FUNCATTN and Attention. All models are trained for 50,000 iterations with a batch size of 8 using the Adam optimizer (Kingma & Ba, 2015). The learning rate is tuned individually for each model.

*Table 8.* Model configurations for sinusoidal regression. All models share parameters between key and query encoders.

|  | **Attention** | **Transolver** | **Intention** | **FUNCATTN** |
|---|---|---|---|---|
| Key/Query Enc. | MLP(3, 256) | MLP(4, 128) | MLP(4, 1000) | MLP(4, 128) |
| Value Enc. | MLP(3, 128) | – | – | – |
| Output Dec. | MLP(2, 128) | – | – | – |
| Heads | 4 | 8 | 8 | 8 |
| Learning Rate | $10^{-3}$ | $10^{-4}$ | $3 \times 10^{-4}$ | $10^{-4}$ |

### C.2. PDE Benchmarks

We benchmark our methods on eight popular PDEs benchmarks across diverse geometries and physical scenarios:

*Table 9.* Summary of benchmark datasets.

| **Benchmark** | **Input** | **Spatial Resolution** | **Input length** | **Output** | **Train/Test** |
|---|---|---|---|---|---|
| Elasticity | Domain geometry | Point cloud | 972 | Displacement $\mathbf{u}$ | 1000/200 |
| Airfoil | Airfoil shape | $221 \times 51$ grid | 11,271 | Density field $\rho$ | 1000/200 |
| Darcy | Permeability | $85 \times 85$ grid | 7,225 | Pressure $u$ | 1000/200 |
| Darcy-Notch | Boundary condition | $51 \times 51$ grid | 2,601 | Pressure $u$ | 1900/100 |
| Pipe | Pipe geometry | $129 \times 129$ grid | 16,641 | Velocity $u_x$ | 1000/200 |
| Navier-Stokes | Vorticity $w_{0:T}$ | $64 \times 64$ grid | 4,096 | Vorticity $w_{T:2T}$ | 1000/200 |
| Plasticity | Punch profile | $101 \times 31 \times T$ | 3,131 | Displacement $\mathbf{u}$ | 900/80 |

**Elasticity (Li et al., 2023a).** This benchmark considers static deformation of a two-dimensional linear elastic body with varying domain geometry, governed by the linear elasticity equations. The input consists of nodal coordinates representing the irregular domain geometry with 972 points per sample, and the output is the corresponding displacement field $\mathbf{g} \in \mathbb{R}^2$ at each node. We use 1000 training and 200 test samples.

**Plasticity (Li et al., 2023a).**    This benchmark simulates dynamic metal forming where an elasto-plastic block obeying $J_2$ plasticity is compressed by a descending rigid punch. The punch profile is generated by interpolating random control points with cubic Hermite splines. Ground truth is computed via finite element analysis on a $101 \times 31$ grid over 20 time steps. The task is to predict the displacement evolution given the punch geometry. We use 900 training and 80 test samples.

**Airfoil (Li et al., 2023a).**    This benchmark studies compressible inviscid flow over a deformable airfoil, governed by the Euler equations. The spatial discretization employs a C-grid mesh with approximately $200 \times 50$ quadrilateral elements. The task is to predict the Mach number field given the mesh point locations as input. We use 1000 training and 200 test samples.

**Pipe (Li et al., 2023a).**    This benchmark studies incompressible viscous flow in a deformable pipe, governed by the incompressible Navier-Stokes equations with viscosity $\nu = 0.005$. A parabolic velocity profile is imposed at the inlet, with free boundary at the outlet and no-slip condition at the pipe surface. The spatial discretization uses a $129 \times 129$ mesh. The task is to predict the horizontal velocity field given the mesh point locations as input. We use 1000 training and 200 test samples.

**Darcy (Li et al., 2021).**    This benchmark models steady-state pressure distribution in heterogeneous porous media, governed by a second-order elliptic PDE on a unit square domain with homogeneous Dirichlet boundary conditions. The task is to learn the nonlinear mapping from the spatially varying permeability field to the pressure head. Solutions are computed on a $421 \times 421$ mesh and subsampled to $85 \times 85$ for training. We use 1000 training and 200 test samples.

**Darcy Flow with Notch in Triangular Domain (Tripura & Chakraborty, 2022)**    This benchmark extends the standard 2D Darcy problem to a more challenging geometric setting, where the flow medium is defined on a triangular domain containing an interior notch. The flow is governed by the Darcy equation with a fixed permeability field $a(x, y) = 0.1$ and forcing function $f(x, y) = -1$. The boundary conditions on the triangular domain are generated using a Gaussian process $u(x) \sim \mathcal{GP}(0, \mathcal{K}(x, x'))$ with kernel $\mathcal{K}(x, x') = \exp\left(-(x - x')^2/2l^2\right)$, where $l = 0.2$ and $x, x' \in [0, 1]$. The task is to learn the operator that maps the boundary conditions to the pressure field over the entire domain. Solutions are computed on a $101 \times 101$ mesh and subsampled to $51 \times 51$ for training. We use 1900 training and 100 test samples.

**Navier-Stokes (Li et al., 2021).**    This benchmark studies incompressible viscous fluid dynamics through the vorticity transport formulation on a periodic unit square domain. We consider the turbulent regime with viscosity $\nu = 10^{-5}$. The spatial discretization uses a $64 \times 64$ grid. Each trajectory consists of 20 temporal snapshots; the task is to predict the latter 10 frames given the initial 10 frames. We use 1000 training and 200 test samples.

**Burgers (Li et al., 2021).**    This benchmark models one-dimensional viscous fluid dynamics governed by the nonlinear Burgers' equation on a periodic domain with viscosity $\nu = 0.1$. The task is to predict the solution at terminal time $t = 1$ given the initial condition, which is sampled from a Gaussian random field. Solutions are computed on a mesh of $2^{13}$ points and subsampled to lower resolutions. We use 1024 training and 100 test samples.

**OOD Generalization AirfRANS.**    The AirfRANS dataset (Bonnet et al., 2022) contains high-fidelity simulation data for Reynolds-Averaged Navier-Stokes (RANS) equations, designed to assist airfoil design. The dataset features airfoils from the NACA 4- and 5-digit series, with each case discretized into approximately 32,000 mesh points. The simulation records air velocity, pressure, and viscosity in the surrounding space, as well as surface pressure. In our experiments, we evaluate on the out-of-distribution (OOD) test splits, specifically the *Scarce* regime for both angle of attack (AoA) and Reynolds number variations. These OOD splits are constructed by holding out samples with extreme parameter values during training, providing a challenging benchmark for assessing the generalization capability of neural surrogate models. Following prior work (Wu et al., 2024), we focus on predicting the surface pressure field, which is essential for estimating lift coefficients relevant to aircraft take-off and landing performance.

**Evaluation metrics.**    We evaluate all methods using the relative $L_2$ error on the test set. Let $g$ denote the ground-truth solution obtained from numerical simulations and $\hat{g} = \mathcal{O}_\theta(f)$ the model prediction. The test error is computed as:

$$\text{Rel. } L_2 = \frac{1}{N_{\text{test}}} \sum_{i=1}^{N_{\text{test}}} \frac{\|g_i - \hat{g}_i\|_{L_2(\Omega)}}{\|g_i\|_{L_2(\Omega)}}, \tag{56}$$

where $\| \cdot \|_{L_2(\Omega)}$ denotes the $L_2$ norm over the spatial or spatial-temporal domain. For training, we minimize the same relative $L_2$ loss on the training set.

Additionally, for AirfRANS, we evaluate the relative $L_2$ error of drag and lift coefficients derived from the predicted physics fields. For unit density fluid, the drag coefficient $C_D$ and lift coefficient $C_L$ are defined as (Wu et al., 2024):

$$C_D, C_L = \frac{2}{v^2 A} \left( \int_{\partial\Omega} p(\boldsymbol{\xi}) \left( \hat{\mathbf{n}}(\boldsymbol{\xi}) \cdot \hat{\mathbf{d}} \right) \mathrm{d}\boldsymbol{\xi} + \int_{\partial\Omega} \tau(\boldsymbol{\xi}) \cdot \hat{\mathbf{d}} \, \mathrm{d}\boldsymbol{\xi} \right), \tag{57}$$

where $\hat{\mathbf{d}}$ is the drag or lift direction respectively, $v$ is the inlet flow speed, $A$ is the reference area, $\partial\Omega$ is the object surface, $p$ is the pressure, $\hat{\mathbf{n}}$ is the outward unit normal, and $\tau$ is the wall shear stress. We also report Spearman's rank correlation $\rho$ (Spearman, 1961) between predicted and ground truth coefficients across test samples, which measures how well the model preserves the ranking of designs—a key property for engineering optimization.

**Training details.** We use a consistent architecture across all benchmarks with 8 transformer layers and 8 attention heads to match previous work. The hidden channel dimension is set to 128 for most benchmarks, while we increase it to 256 for Navier-Stokes and AirfRANS due to their higher complexity. The number of bases is set to 64 for standard benchmarks and reduced to 32 for Navier-Stokes and AirfRANS to balance computational cost and expressiveness. We further found that sharing the learnable basis modules across layers encourages the model to learn more structured bases, which improves accuracy.

*Table 10.* Training and model configurations for FUNCATTN. Training configurations follow prior works (Hao et al., 2023; Wu et al., 2024) without extra tuning. $\mathcal{L}_g$ denotes spatial gradient regularization (Xiao et al., 2024). $\mathcal{L}_v$ and $\mathcal{L}_s$ denote volume and surface losses respectively.

| | Training Configuration | | | | | Model Configuration | | | |
| Benchmark | Loss | Epochs | LR | Optim | Batch | Layers | Heads | Channels | Modes |
|---|---|---|---|---|---|---|---|---|---|
| Elasticity | | | | | 1 | 8 | 8 | 128 | 64 |
| Plasticity | | | | | 8 | 8 | 8 | 128 | 64 |
| Airfoil | Rel. $L_2$ | 500 | $10^{-3}$ | AdamW | 4 | 8 | 8 | 128 | 64 |
| Pipe | | | | | 4 | 8 | 8 | 128 | 64 |
| Navier-Stokes | | | | | 2 | 8 | 8 | 256 | 32 |
| Darcy w/ Notch | | | | | 25 | 8 | 8 | 128 | 64 |
| Darcy | Rel. $L_2 + 0.1\mathcal{L}_g$ | | | | 4 | 8 | 8 | 128 | 64 |
| AirfRANS | $\mathcal{L}_v + \mathcal{L}_s$ | 400 | $10^{-3}$ | Adam | 1 | 8 | 8 | 256 | 32 |

# D. Ablation

## D.1. Number of Basis

Table 11 presents the complete ablation study on the number of bases. As discussed in Section 5.7, moderate mode counts (64–128) achieve the best balance between expressiveness and generalization. Notably, the optimal number of modes varies across tasks: Elasticity and Plasticity favor 256 bases, while Darcy benefits from higher counts. The optimal mode count varies by task, likely reflecting differences in solution smoothness across PDE systems.

## D.2. Transpose vs. Pseudo-Inverse Projection

As motivated in Remark 4.1, we use the transpose $\boldsymbol{\Phi}^\top$ in place of the Moore–Penrose pseudo-inverse $\boldsymbol{\Phi}^\dagger = (\boldsymbol{\Phi}^\top \boldsymbol{\Phi})^{-1} \boldsymbol{\Phi}^\top$. The unregularized pseudo-inverse causes exploding gradients in our experiments. A Tikhonov-stabilized variant $\boldsymbol{\Phi}^\dagger_\lambda = (\boldsymbol{\Phi}^\top \boldsymbol{\Phi} + \lambda \mathbf{I}_k)^{-1} \boldsymbol{\Phi}^\top$ (Hoerl & Kennard, 1970) resolves this, but introduces an additional regularizer and increases the condition number of the inverted matrix in Eq. (8) by more than an order of magnitude, as shown in Figure 6. In contrast, the transpose yields stable training, lower computational cost, and better accuracy, reported in Table 12.

## D.3. Sensitivity to Tikhonov Parameter $\lambda$

Remark 4.2 and Proposition 4.5 both suggest that the Tikhonov term $\lambda \|\mathbf{C}\|_F^2$ in Eq. (6) primarily serves to stabilize the linear solve. Here we verify this empirically. In our implementation, $\lambda = \mathrm{sigmoid}(\alpha)$ is learnable through a scalar $\alpha$,

*Table 11.* Ablation study on the number of bases. We report relative $L^2$ error (%) across six benchmark tasks. Inference time on Elasticity (ms/sample) and peak GPU memory (GB) are also reported using Nvidia A2000.

| Modes | Relative $L^2$ Error (%) ↓ | | | | | | Computational Cost | |
|---|---|---|---|---|---|---|---|---|
| | **Elasticity** | **Plasticity** | **Airfoil** | **Pipe** | **NS** | **Darcy** | **Time** ↓ | **Memory** ↓ |
| 16 | 0.65 | 0.12 | 0.51 | 0.30 | 13.53 | 0.49 | 12.52 | 0.02 |
| 32 | 0.55 | 0.13 | 0.52 | 0.31 | 8.09 | 0.45 | 13.28 | 0.02 |
| 64 | 0.50 | 0.11 | 0.43 | 0.29 | 8.00 | 0.42 | 13.65 | 0.02 |
| 128 | 0.49 | 0.13 | **0.42** | **0.27** | **7.82** | 0.44 | 16.35 | 0.02 |
| 256 | **0.48** | **0.10** | 0.47 | 0.29 | 8.15 | 0.43 | 34.60 | 0.04 |
| 512 | 0.56 | 0.13 | 0.48 | 0.35 | 8.32 | **0.41** | 75.48 | 0.09 |

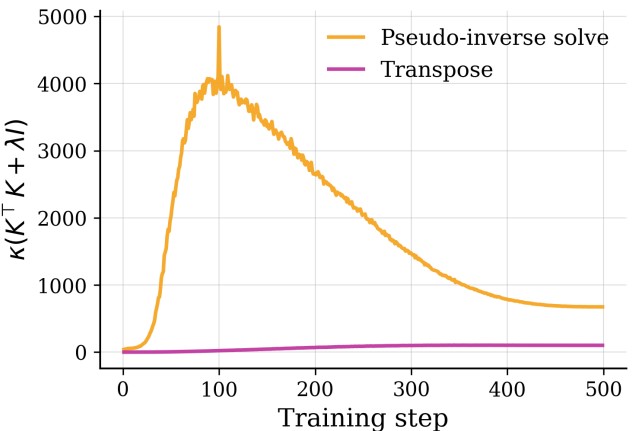

*Figure 6.* Condition number of the inverted matrix in Eq. (8) during training on Elasticity, comparing the Tikhonov-stabilized pseudo-inverse and the transpose.

*Table 12.* Test error (relative $L_2$, $\times 100$) on Elasticity and Darcy for the two projection choices.

| Projection | Elasticity | Darcy |
|---|---|---|
| Stabilized pseudo-inverse | 0.51 | 0.44 |
| Transpose | **0.50** | **0.42** |

and we vary its initialization $\alpha_{\text{init}}$ to study how the strength of regularization affects training. Figure 7 tracks the average condition number $\kappa(\tilde{\mathbf{K}}\tilde{\mathbf{K}}^\top + \lambda \mathbf{I}_k)$ across the 8 FUNCATTN layers throughout training on Elasticity.

Smaller $\alpha_{\text{init}}$ corresponds to weaker regularization and raises the final condition number from $\sim 8$ to $\sim 100$, yet the test error on Elasticity varies by less than $0.02$ across the three settings, as shown in Table 13. Within this range, FUNCATTN is thus robust to the choice of $\lambda$, with relaxed regularization even yielding a mild accuracy gain. Combined with the instability observed when $\lambda \to 0$ in Remark 4.2 and the $1/\lambda^2$ scaling in Proposition 4.5, this suggests that $\lambda$ acts as a numerical safeguard: it must remain strictly positive, but its exact value is not a sensitive hyperparameter.

## E. Visualization

### E.1. Basis Visualization

As shown in Fig. 8, we visualize the learned basis functions for different models. FUNCATTN learns smooth, localized bases that capture regional features. In contrast, Transolver produces highly sparse activations concentrated at scattered points, which may limit its ability to represent smooth solution fields. When we impose orthogonality constraints (Fig. 8c), the bases become globally supported and resemble Fourier modes, suggesting that explicit regularization encourages the model to recover classical spectral structure. We hypothesize that strict orthogonality over-regularizes the representation, preventing the model from capturing task-specific structure.

### E.2. PDE Visualization

We provide qualitative comparisons between FUNCATTN and Transolver across all six benchmarks in Figs. 11–12. We visualize absolute error maps ($|\hat{g} - g|$) to highlight spatial error distributions, complementing the scalar relative $L_2$ metrics

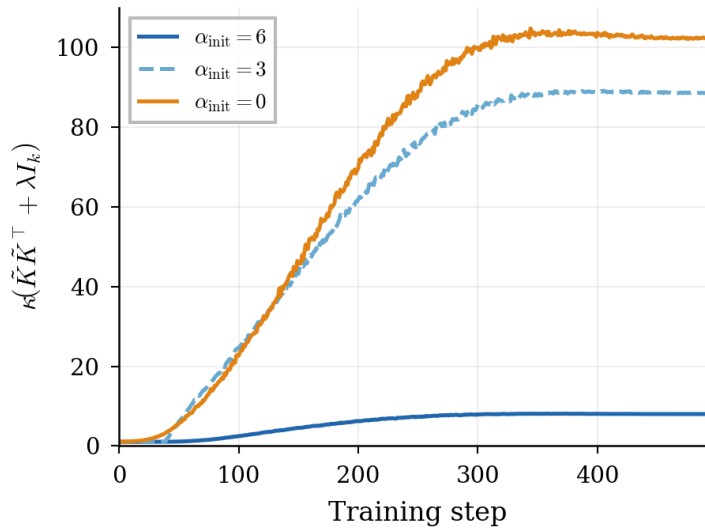

*Table 13.* Final condition number $\kappa$ and test error (relative $L_2$, $\times 100$) on Elasticity for different initializations of $\alpha$.

| $\alpha_{\text{init}}$ | $\kappa$ **(final)** | **Test Error** |
|---|---|---|
| 0 | $\sim 100$ | **0.48** |
| 3 | $\sim 90$ | 0.49 |
| 6 | $\sim 8$ | 0.50 |

*Figure 7.* Average condition number of the inverted matrix in Eq. (8) during training on Elasticity, for different initializations of $\alpha$.

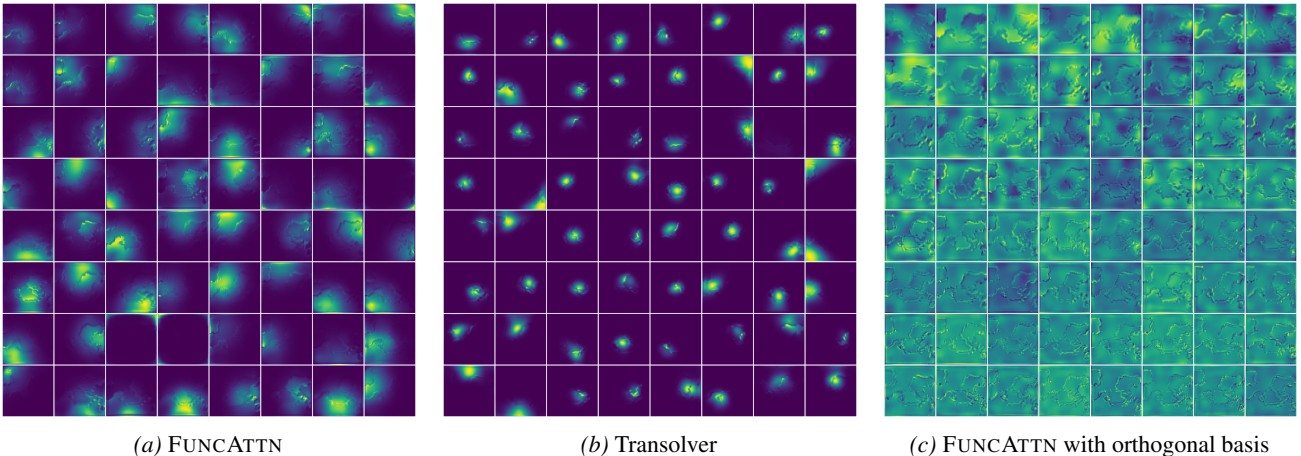

| *(a)* FUNCATTN | *(b)* Transolver | *(c)* FUNCATTN with orthogonal basis |
|---|---|---|

*Figure 8.* Visualization of learned basis for different models.

in the main text. This reveals where each model struggles, such as near boundaries or in regions with sharp gradients.

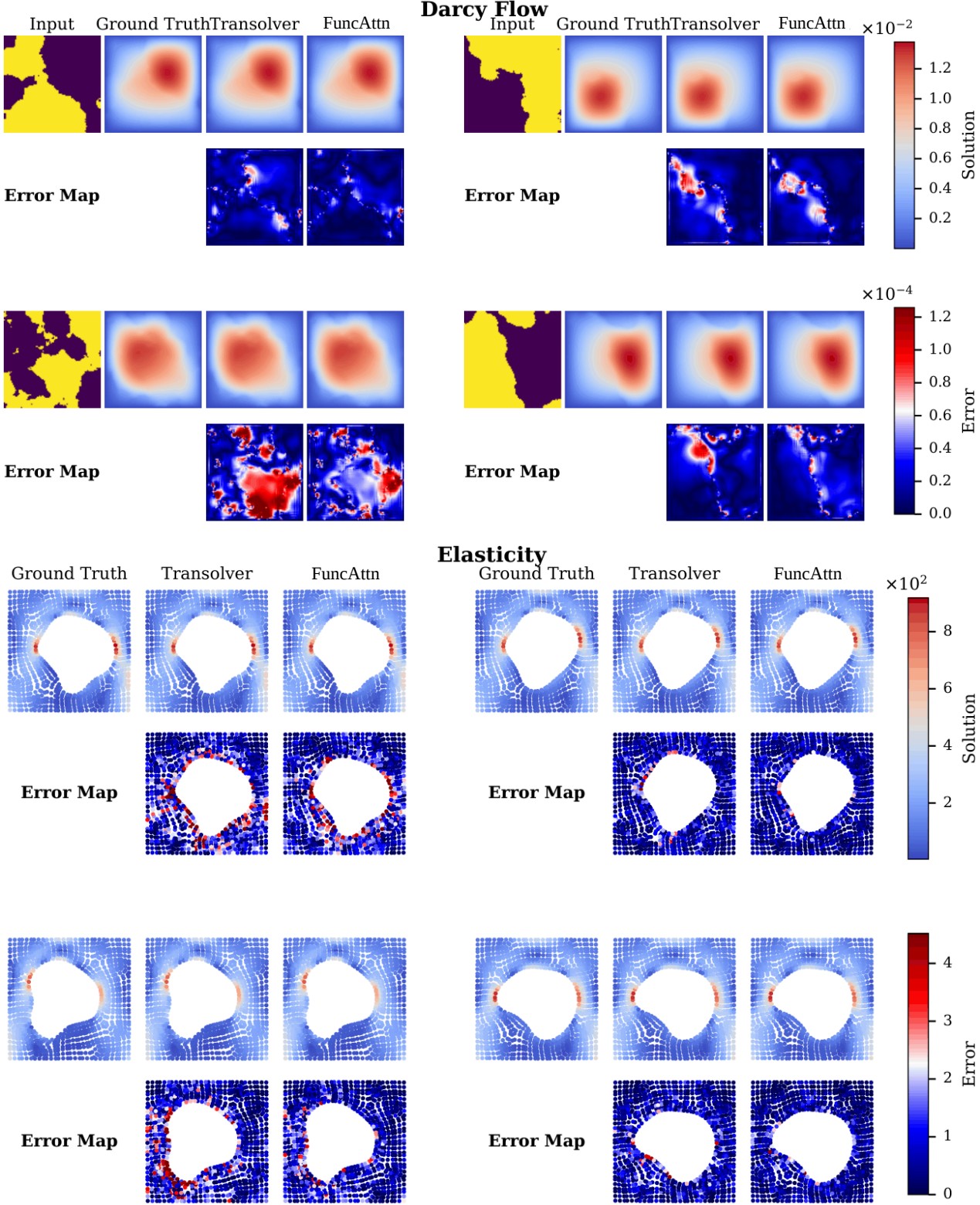

*Figure 9.* **Prediction Visualizations.** (Top) Darcy flow solution fields. (Bottom) Elasticity stress fields on irregular meshes. Each shows ground truth, Transolver, and FUNCATTN with error maps.

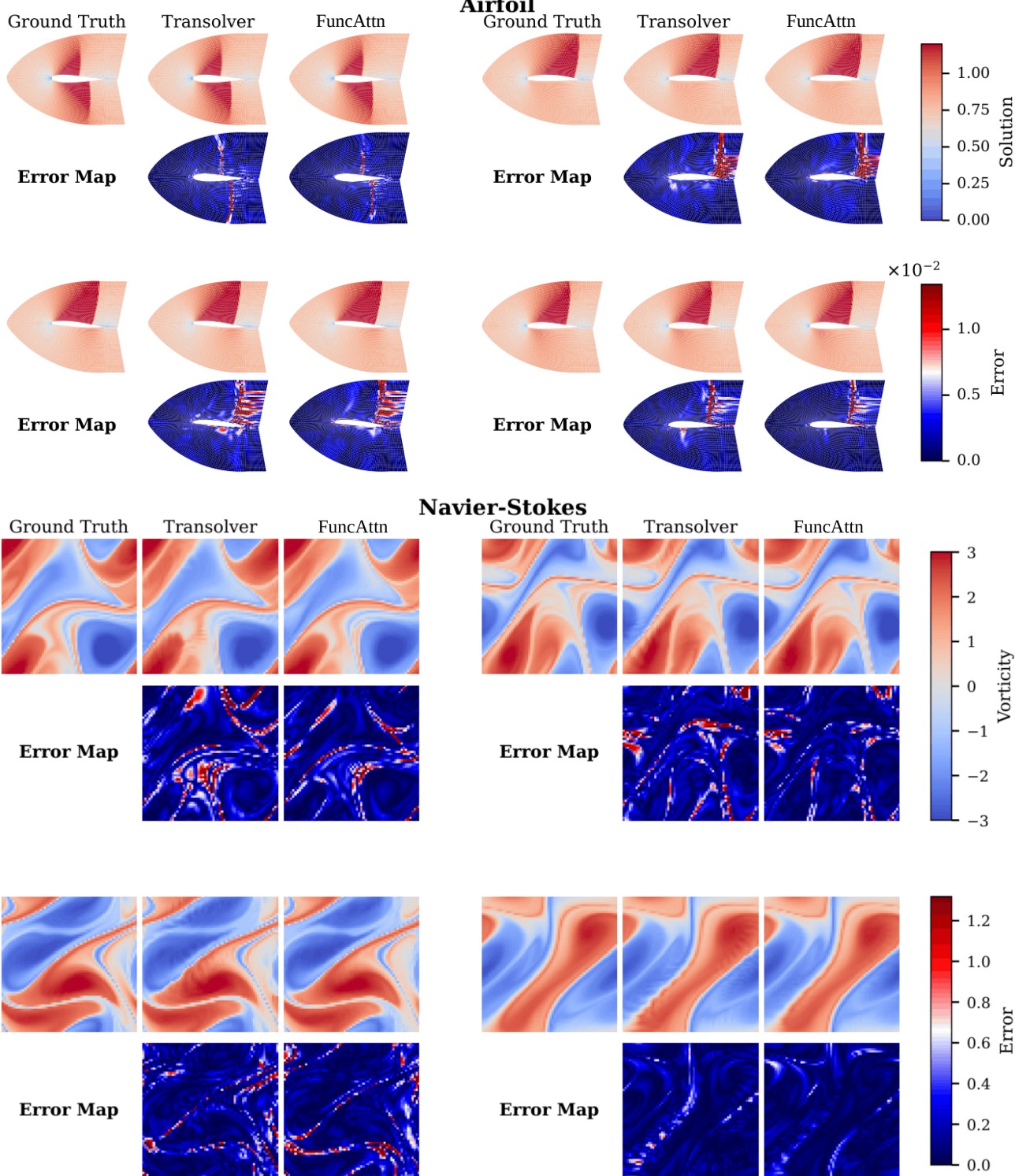

*Figure 10.* **Prediction Visualizations.** (Top) Airfoil velocity fields. (Bottom) Navier-Stokes vorticity fields at $t = 20$ after rollout.

## Plasticity

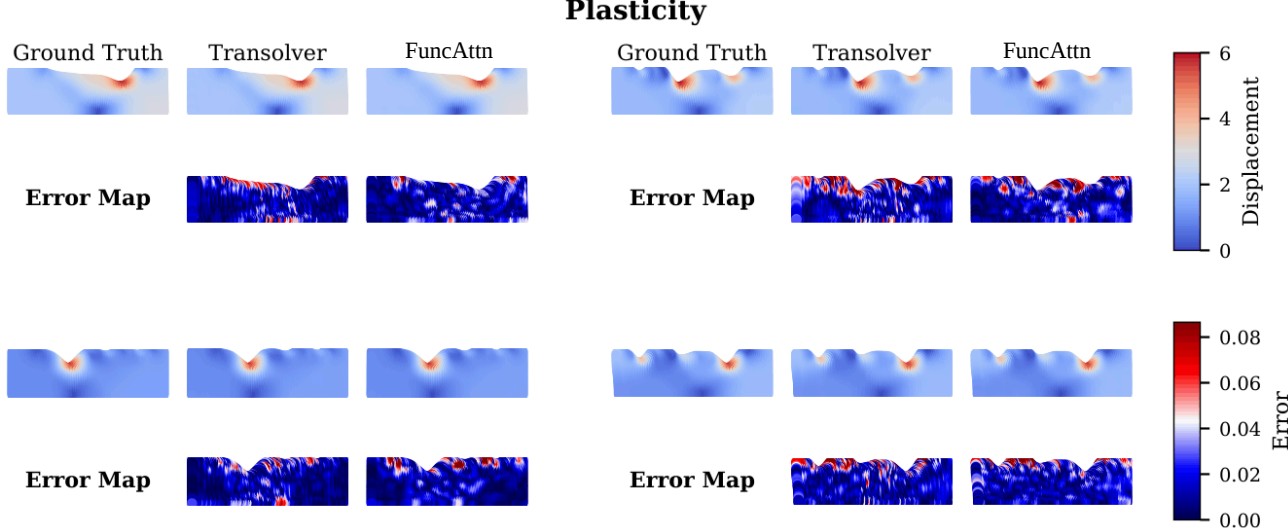

*Figure 11.* **Prediction Visualizations.** Plasticity displacement magnitude fields at the final timestep.

## Pipe Flow

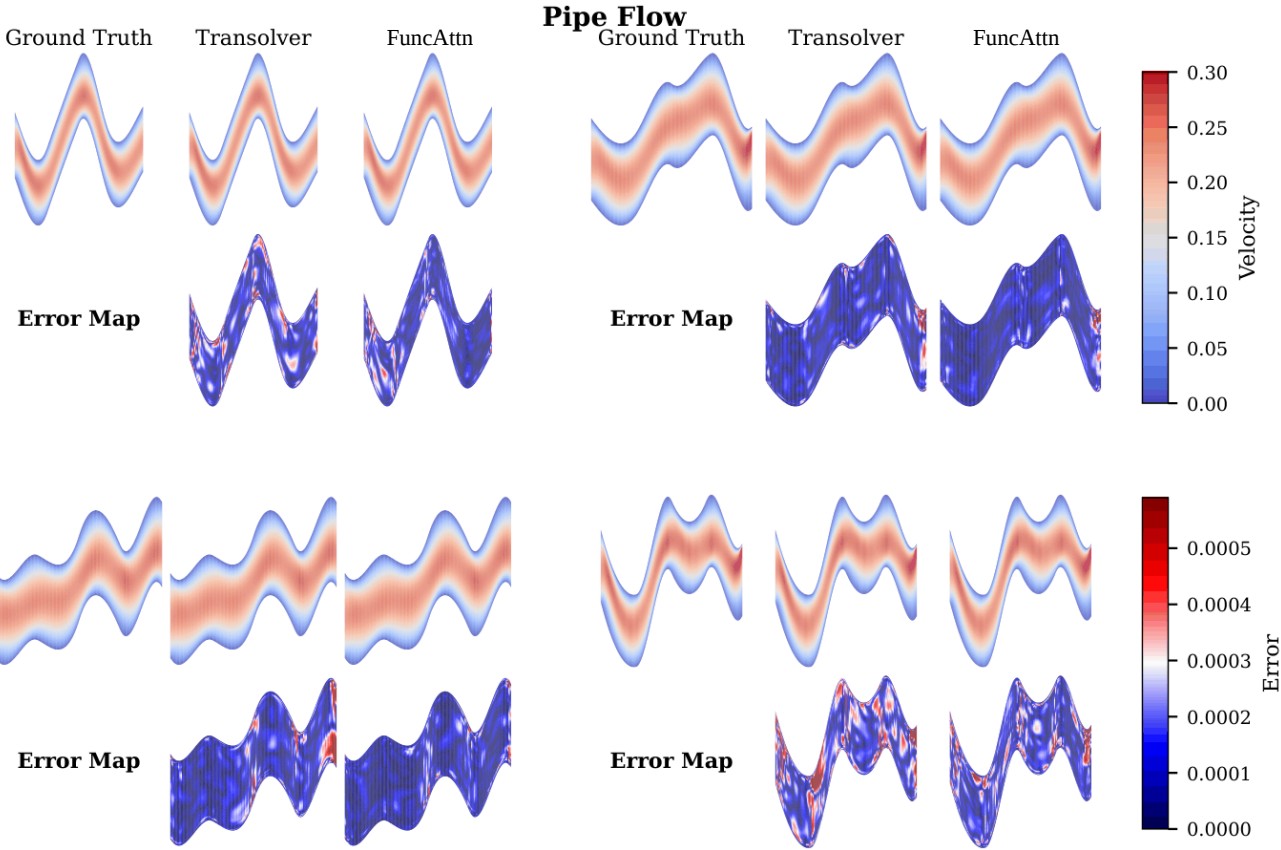

*Figure 12.* **Prediction Visualizations.** Pipe flow velocity fields on irregular meshes.

