# OpenReview forum: "Functional Attention: From Pairwise Affinities to Functional Correspondences"
_ICML.cc/2026/Conference — ICML 2026 regular_

### Official Review · Reviewer_4Cdf · 2026-02-18

**Soundness:** 3
**Presentation:** 3
**Significance:** 3
**Originality:** 3
**Overall Recommendation:** 5
**Confidence:** 3

**Summary:**

This paper revisits the theoretical basis of attention in Transformers and argues that viewing attention solely as a matrix of pairwise token similarities is overly narrow. Instead, it presents a broader formulation in which attention is interpreted as a functional correspondence, namely an operator that acts between representations at the level of functions rather than only through token-to-token interactions. By drawing connections to kernel methods and operator theory, the paper develops a more global and mathematically grounded account of attention. It also outlines how this perspective may be relevant for structured or continuous data and places attention within a wider functional-analytic framework.

**Compliance With Llm Reviewing Policy:**

Affirmed.

**Final Justification:**

Recommended to accept

**Key Questions For Authors:**

1.It would strengthen the paper to provide a more detailed comparison of runtime and memory consumption among FUNCATTN, standard softmax attention, and recent low-rank or linear attention variants over a range of sequence lengths 𝑛. Clear experimental settings and scaling plots would make the efficiency discussion much more convincing. It would also be useful to clarify whether the method encounters any practical bottlenecks in speed or memory at larger scales.

2.The cardinality parameter 𝑘 appears to be a key design factor in the method. More systematic analysis of how performance and generalization change when 𝑘 is set to very small or very large values would improve the paper. Some practical guidance on how to choose 𝑘 across different tasks or domains would also make the approach easier to use.

3.The current evaluation seems to focus mostly on relatively regular domains. It would be valuable to examine performance in more challenging settings, such as non-uniform sampling patterns or domains with strong local singularities. Even if additional experiments are not feasible, a discussion of expected behavior and likely failure cases in these settings would provide helpful context.

4.The paper would also benefit from clearer positioning with respect to recent Transformer-based architectures for operator learning. Expanding this discussion, and including further comparisons where possible, would help clarify how FUNCATTN differs in its handling of global and local dependencies and in what settings it offers the most meaningful advantages.

**Limitations:**

No. The paper mainly concentrates on the theoretical perspective and methodological formulation, while its discussion of limitations and broader societal implications remains fairly brief. It would improve the paper to state more explicitly that the contribution is still largely conceptual and that issues such as large-scale performance, stability, and scalability have not yet been comprehensively established. It would also be useful to discuss potential limitations related to computational overhead, sensitivity to important hyperparameters, and robustness in irregular or pathological settings. Although the work does not raise an immediately obvious harmful-use concern, advances in operator learning or large-scale modeling could still carry indirect consequences in high-impact application domains. A short and balanced discussion along these lines would make the paper more transparent and more complete.

**Strengths And Weaknesses:**

Strengths

1.A notable strength of the paper is its attempt to rethink attention at a more abstract level. Rather than staying with the standard pairwise similarity interpretation, it frames attention as a form of functional correspondence, which gives the discussion a broader and more structured perspective.

2.The paper also benefits from linking its argument to established mathematical tools, especially kernel methods and operator theory. This connection helps place Transformers in a wider theoretical setting and makes the proposed viewpoint more intellectually grounded.

3.The operator-based viewpoint suggests potential extensions to continuous domains, geometric data, and other structured settings.

4.The presentation is generally clear and well organized. The paper moves in a logical way from motivation to formal discussion, which makes the main conceptual contribution relatively easy to follow.

Weaknesses

1.The main limitation is that the contribution remains primarily conceptual. The paper does not offer new theorems, formal guarantees, or detailed analysis of expressivity and generalization, so the theoretical contribution feels somewhat limited in depth.

2.The novelty would also benefit from sharper positioning against prior work. Since links between attention and kernel-based interpretations have already appeared in the literature, the paper should explain more clearly what is genuinely new in its perspective.

3.Empirical support is still fairly limited. At present, it is not fully clear whether this functional view translates into better architectures, stronger algorithms, or measurable gains in practical settings.

4.In a related sense, the paper does not yet produce a clear algorithmic contribution that directly follows from the proposed reinterpretation. This weakens the practical significance of the work, even if the conceptual framing is interesting.

---

> ### Author Rebuttal · Authors · 2026-03-30
>
> Thanks for your thoughtful feedback and appreciation of our efforts on "a global and mathematically grounded account of attention", which "helps place Transformers in a wider theoretical setting and makes the proposed viewpoint more intellectually grounded". Below we address your questions.
> # Q1: Runtime, Memory Analysis
> We conducted a scaling experiment comparing FuncAttn against softmax attention, Performer, Linformer, Nyströmformer over $n \in \{256, \dotsc, 16384\}$ with $d{=}128$, $h{=}8$, $k{=}64$, batch size 1, on an NVIDIA A2000. (see https://anonymous.4open.science/r/Figures-8D4F/scaling_runtime_memory.pdf). Both runtime and memory scale linearly in $n$, consistent with $O(ndk)$ (Appendix B). We will include this figure in the final manuscript.
>
> **Practical bottleneck.** At large $n$, the dominant cost is basis projection with rate $O(ndk)$. Memory grows linearly via the $n \times k$ basis matrix. Please refer to Q4 of 5pVm for a discussion on runtime.
>
> # Q2: Ablation on k
> We appreciate this suggestion. Our ablation (Tab 10) reports $k \in \{16, 32, 64, 128, 256, 512\}$ across all six benchmarks. We summarize the key patterns and practical guidance:
>
> **Extreme regimes.** When $k$ is too small ($k{=}16$), the basis cannot adequately resolve the operator, this is most pronounced on spatially complex tasks like Elasticity (0.65 vs. 0.50 at $k{=}64$) and Navier-Stokes (13.53 vs. 8.00). On the other end, pushing $k$ to 512 does not yield meaningful gains on most tasks (e.g., Elasticity worsens to 0.56), yet incurs a ${\sim}5.5\times$ wall-clock overhead (75.48 ms vs. 13.65 ms per sample, Tab 10).   Please refer to Q4 of 5pVm for a discussion of k on runtime.
>
> **Practical advice.** In practice, $k{=}64$ works well as a default, it sits within ${\sim}5\%$ of the best result on every benchmark we tested, at negligible extra cost over smaller values. The right $k$ does depend on the problem: for smoother target fields like Darcy and Pipe, $k{=}32$–$64$ is already sufficient, whereas tasks with finer spatial structure (Elasticity, Navier-Stokes) see continued gains up to $k{=}128$ or $256$. Our recommendation is to start with $k{=}64$ and increase to $128$ if the target field is known to exhibit multiscale or high-frequency features.
>
> # Q3: More Challenging Setting
> Several evaluated benchmarks already involve complex geometric settings: Elasticity uses irregular point clouds, and Airfoil uses non-uniform meshes. To further challenge our methods, we conduct an additional experiment on the 2D Darcy flow on a triangular domain with a geometric notch [1], where the domain is non-rectangular, the notch tip introduces a geometric singularity, and the mesh is non-uniform.
>
> **Tab A:** Mean $L^2$ relative error (%) on 2D Darcy flow with triangular notch domain. Results for baselines are from [1, Tab 3];
>
> | **Method** | **Rel. $L^2$ Error (%)** |
> |:---|:---|
> | DeepONet | $2.64$ |
> | POD-DeepONet | $1.00$ |
> | MWT | $0.87$ |
> | dgFNO+ | $7.82$ |
> | WNO (reported) | $0.77$ |
> | WNO (reproduced) | $0.92$ |
> | **FuncAttn (Ours)** | $\mathbf{0.64}$ |
>
> FuncAttn achieves a 30.9% relative improvement over WNO, which was specifically designed for complex-geometry tasks, further highlighting its strong performance on problems with non-regular domains.
>
> # Q4: Comparison to Transformer-Based Operator Learning Methods
>
> We thank the reviewer for raising this point. Compared to methods like Transolver, our approach starts from a fundamentally different perspective. Transolver begins from the Transformer architecture and introduces task-specific modifications to adapt attention for PDE tasks. These choices are effective but largely guided by empirical intuition.
> FuncAttn begins from a mathematical question: how should one construct correspondences between function spaces? Through this optimization formulation, every component in our method carries a concrete meaning. The number of bases k corresponds to spectral truncation, and the regularization connects to Tikhonov theory. Our ablation studies on basis type and k (Sec 5, Tab 5 and 6) verify that these theoretical insights lead to consistent performance differences across benchmarks.
>
> [1] Tripura & Chakraborty, "Wavelet Neural Operator for Parametric PDEs"

---

> > ### Author Rebuttal · Reviewer_4Cdf · 2026-04-01
> >
> > Thank you for the clarifications. My concerns have been sufficiently addressed, and I will increase my score accordingly.

---

> > > ### Author Response · Authors · 2026-04-07
> > >
> > > We would like to thank Reviewer 4Cdf for your insightful review and for taking the time to re-evaluate our responses. We greatly appreciate your valuable feedback, which has helped us strengthen the paper. The corresponding revisions will be incorporated into the final version.

---

### Official Review · Reviewer_BfSd · 2026-02-22

**Soundness:** 3
**Presentation:** 4
**Significance:** 3
**Originality:** 2
**Overall Recommendation:** 4
**Confidence:** 2

**Summary:**

The paper proposes a new attention formulation for operator learning by interpreting attention as a linear operator between function spaces. The proposed method achieves strong performance across multiple tasks, including PDE solving, 3D segmentation, and regression, and demonstrates improved robustness and generalization compared to existing neural operator and transformer-based approaches.

**Compliance With Llm Reviewing Policy:**

Affirmed.

**Final Justification:**

I would keep 4 score (weak accept), since I understand the novelity in the design of attention mechanisms suitable for operator learning.

**Key Questions For Authors:**

Please answer questions in Major Concerns 1, 2, and 3.

**Limitations:**

Yes.

**Strengths And Weaknesses:**

The paper is technically sound, well organized, and easy to read. It addresses an important problem, namely the design of attention mechanisms suitable for operator learning. The experimental results are very strong, and the proposed method consistently outperforms existing neural operator and transformer-based methods across multiple benchmarks.
However, regarding the relationship to existing work, I have the following major concerns, which are related to the originality of this paper.



(Major Concern 1):

The proposed method is conceptually very close to existing neural operator architectures, particularly low-rank neural operators.
In other words, it seems that the proposed method can be interpreted as an alternative parameterization of existing low-rank neural operators, which raises the question of whether the proposed method represents a fundamentally different class of operators, or simply an alternative parameterization or implementation of existing low-rank neural operators.
Therefore, it would be important for the authors to clarify whether the proposed method provides a new operator class, or whether it is primarily a different parameterization of existing neural operator frameworks.



(Major Concern 2):

The paper argues that existing methods "ignore the global functional structure." However, this claim may be somewhat overstated.
Many modern operator transformers (e.g., OFormer, GNOT, and related models) do not operate purely on function values alone, but rather on joint representations of spatial position and function value, i.e., $(x_i,f(x_i))$ , as the input, and employ positional encodings to incorporate geometric information of spatial positions.
Therefore, the global functional structure is already incorporated into the representation in these transformer models.
Given this, it would be helpful if the authors could more clearly explain whether the proposed approach offers fundamentally new capabilities beyond simply introducing a continuous basis representation, or whether it can capture structural properties that cannot be represented by existing coordinate-based attention mechanisms.




(Major Concern 3):

The experimental results consistently show that the proposed method outperforms existing baselines across multiple benchmarks.
However, in light of the concerns above, it would be valuable to provide further conceptual explanation of why the proposed method achieves superior performance.

---

> ### Author Rebuttal · Authors · 2026-03-30
>
> We are delighted that you acknowledge that “our paper is technically sound, well organized, and easy to read”. Particularly, that “our proposed method achieves strong performance across multiple tasks, … and demonstrates improved robustness and generalization…”. Below we address the concerns raised in the review.
>
> # Q1: Novelty of FuncAttn and its Relation to Existing Low-Rank Neural Operator Methods
> While FuncAttn conceptually aligns with the family of low-rank attention mechanisms, it represents a shift in design. Rather than merely approximating the standard scaled dot-product attention (Sec. 3.2 in our paper), FuncAttn reformulates the attention mechanism through a functional transport method (see Sec. 3.3 "Motivation" in our paper). By leveraging a principled and highly expressive low-rank functional space, our approach demonstrates superior empirical performance across a diverse range of neural operator learning tasks.
>
> Contemporary low-rank methods are primarily engineered to approximate existing attention structures to gain efficiency. F.e., Linformer maintains the vanilla softmax bottleneck while projecting the K and V matrices into a lower-dimensional subspace. Similarly, Nyströmformer employs the Nyström method to linearize the quadratic complexity of the softmax operation. In both cases, the objective is fidelity, striking a balance between computational speed and how closely the model can mimic vanilla attention.
>
> In contrast, FuncAttn introduces a novel geometric perspective on attention design. We propose, for the first time, a linear solver (Eq. 7) operating within a learned spectral space to capture complex spatial relations. By bridging attention mechanisms with the well-established mathematical framework of functional maps used in geometry processing and shape analysis, FuncAttn is conceptually novel and differs from existing low-rank methods.
>
> # Q2: What does FuncAttn offer beyond positional encoding?
> While positional encodings inject spatial coordinates into individual tokens, thereby incorporating the global structure necessary for tasks such as solving the Navier-Stokes equations, they face significant challenges in the presence of non-rigid deformations. In domains like 3D RNA segmentation, the key information resides in the intrinsic structure of the data rather than its extrinsic spatial or relative coordinates (see Tab 3).
>
> Furthermore, even with positional information, these models still rely on computationally expensive pairwise token-level affinities. This dependency makes such methods highly sensitive to the sampling density of the underlying function, failing to exploit the lower-dimensional functional relationships that may exist within the data.
>
> In contrast to relying on fixed positional encodings, FuncAttn captures global geometric and physical structures by learning adaptive bases. By operating inherently within a learned spectral domain, our approach achieves both global context awareness and computational efficiency simultaneously. This shift from coordinate-based attention to basis-driven functional maps allows the model to remain invariant to sampling density while maintaining high fidelity to the underlying physical manifold (Sec 5.5) .
>
> # Q3: Insights of FuncAttn's superior performance
>
> Our superior performance is primarily driven by two core design innovations, besides the insights discussed in Sec 3.3:
>
> **1. Input-Adaptive Transport**: Unlike established models such as FNO and LNO, which employ fixed kernels within their latent bottlenecks, FuncAttn utilizes an input-dependent transport operator. Specifically, the matrix C\* = Q̃K̃ᵀ(K̃K̃ᵀ + λIₖ)⁻¹ adapts dynamically to the local solution geometry of each input. This allows the model to respond to the specific geometric features of the data rather than applying a static transformation. Crucially, our ablation studies reveal that even when constrained to fixed Fourier bases, FuncAttn continues to outperform all baselines. This confirms that the efficiency of the transport mechanism provides a significant performance gain (Tab 6).
>
> **2. Learned Non-Orthogonal Partition-of-Unity Bases**: Empirical results (Tab 6) demonstrate that freely learned bases consistently outperform both Fourier-based and orthogonally-constrained alternatives across all tested attention mechanisms. By employing a softmax-normalized basis(Proposition 4.1), we ensure the representation remains bounded and satisfies the partition-of-unity property. This prevents degenerate compression while maintaining high geometric adaptivity (see Fig. 5a and 5b).
>
> These design choices coalesce into a principled and generalized attention formalism viewed through a functional transport. This framework strikes a balance between architectural flexibility and structural informativeness. Furthermore, we demonstrate the extensibility of this approach by showing that it recovers popular architectures, such as IntentionNet, as special cases (see A4).

---

> > ### Author Rebuttal · Reviewer_BfSd · 2026-04-01
> >
> > Thank you for the clarification. I now better understand the originality of this paper. I would like to maintain my score.

---

> > > ### Author Response · Authors · 2026-04-07
> > >
> > > We are happy to hear that your concerns have been fully addressed, and we appreciate your thoughtful follow-up. Your feedback throughout the review process has helped us improve the clarity of our work.

---

### Official Review · Reviewer_qVXE · 2026-03-11

**Soundness:** 2
**Presentation:** 3
**Significance:** 2
**Originality:** 3
**Overall Recommendation:** 4
**Confidence:** 4

**Summary:**

This paper introduces "Functional Attention" (FUNCATTN), a reformulation of the standard attention mechanism for operator learning. Normally, transformers treat continuous physical fields as lists of independent, discrete tokens, calculating a big $n \\times n$ point-wise similarity matrix. This is computationally expensive and also can miss the broader, global structure of the data.

To fix this, the authors take inspiration from geometric functional maps, which is a soft shape correspondence tool from graphics. Instead of relying on a standard softmax dot-product, they project the queries, keys, and values into a much smaller, learned spectral space using adaptive bases. To find the alignment between query and key-value, they set up a regularized least-squares regression and directly solve for the mathematically optimal $k \times k$ linear transport matrix, $C$, to align the keys and queries.

By doing this in a reduced space, the model inherently smooths out high-frequency noise and aggregates broader regional context. The core contributions are this theoretical framing of attention as a soft functional correspondence, alongside strong empirical results. The authors demonstrate that FUNCATTN achieves state-of-the-art performance across continuous tasks—like 1D few-shot regression, 2D/3D PDE solving, and 3D RNA segmentation—while generalizin well to unseen grid resolutions and out-of-distribution physical parameters.

**Compliance With Llm Reviewing Policy:**

Affirmed.

**Key Questions For Authors:**

**Q1. Theoretical Convergence and Approximation Bounds (Ref W1):** Could you provide a theoretical sketch or analysis of a single FUNCATTN layer? Specifically, assuming input functions in a standard Hilbert or Sobolev space, does the joint optimization of the softmax bases and the transport matrix $C$ asymptotically converge to the span of known optimal bases as $k$ increases? Most importantly, what is the formal approximation error bound between the true operator and your predicted operator?

**Q2. Truncation Error and Non-Smooth Data (Ref W2 & W3):** Since mapping to a $k$-dimensional space acts as a low-pass filter, is there a formal theoretical bound on the truncation error introduced by this implicit low-rank constraint? Furthermore, how does your functional correspondence mathematically behave (or break down) when the underlying signal lacks fast spectral decay, such as in the presence of sharp shocks or sudden spatial transitions?

**Q3. Numerical Robustness and Matrix Conditioning (Ref W1 & W4):** The closed-form solution for $C$ requires inverting a regularized spectral matrix. Since the learned bases are unconstrained and can easily become collinear, how does the condition number of this matrix behave during training? Have you formally analyzed the Lipschitz continuity of the learned operator under bounded input perturbations? Finally, how sensitive is the model's stability and performance to the specific choice of the Tikhonov regularization parameter $\lambda$?

I find the functional alignment idea and the empirical results very interesting in the paper. However, my current assessment of the paper's Soundness is held back by the lack of formal mathematical guarantees, which are standard for operator learning theory. If the rebuttal can provide concrete theoretical sketches for the approximation error bounds (Q1 & Q2) and a formal mathematical justification of the numerical stability and conditioning (Q3), I would be willing to raise my Soundness and Significance scores.

**Limitations:**

yes

**Strengths And Weaknesses:**

**Strengths**

**S1: A new perspective.** The core idea of reframing attention as a functional correspondence in the paper is genuinely new. Taking the concept of functional maps from 3D geometry processing and non-rigid shape matching and using it to replace the standard softmax-based $n \times n$ affinity matrix in neural networks is an innovative cross-pollination of ideas. Instead of forcing point-wise token matches, solving for the optimal $k \times k$ linear transport matrix in a learned space gives a much more mathematically grounded way to handle continuous physical fields.

**S2: Extensive and convincing empirical validation.** The experimental setup is thorough. The authors tested the method across 1D few-shot regression, various 2D/3D PDE benchmarks (like Navier-Stokes and Elasticity), and 3D point cloud segmentation. It is good to see that it works well on out-of-distribution physical parameters and zero-shot super-resolution, which validates the learning of discretization-independent operators as claimed in the paper.

**S3: Overcoming quadratic complexity.** Moving the attention calculation into a reduced $k \times k$ space is shown to sidestep the quadratic scaling bottleneck of standard transformers. By projecting the queries and keys into a compact spectral space first, the actual alignment solve—the matrix inversion—becomes independent of the sequence length $n$. This makes the model much more scalable for high-resolution grid data or dense point clouds where traditional attention would be computationally inefficient.

**Weaknesses**

**W1:** The paper presents an effective way to improve attention calculation, but it lacks deep theoretical analysis, which could be done given that functional map theory is well-established in geometry processing and graphics. Analyzing a full deep network is certainly intractable, but it would be highly beneficial to analyze a single layer involving the joint learning of basis functions and the closed-form functional alignment. For example, if the input functions belong to a standard Hilbert or Sobolev space, does the joint optimization of the softmax bases and the transport matrix $C$ asymptotically converge to the span of known optimal bases (like Laplacian eigenfunctions) as the basis count $k$ increases? Furthermore, what would be the theoretical approximation error bound between the true operator and the predicted operator under that norm? Finally, the paper lacks a formal stability analysis. If we would inject a bounded perturbation into the input field, would the learned functional operator be Lipschitz continuous? It would be also great to know how the condition number of the inverted spectral matrix behaves under noisy or adversarial data, as a poorly conditioned matrix could cause the least-squares solve to become numerically unstable.

**W2:** The paper achieves efficiency by mapping to a reduced $k$-dimensional space, which would naturally acts as a low-pass filter and introduces over-smoothing. The authors rightly noted this acts as an implicit low-rank constraint. However, the paper does not theoretically quantify this effect. Is there a formal bound on the truncation error introduced by this constraint? This would strengthen the paper to formally compare this smoothing effect and its spectral bias against standard low-rank approximation methods (like Nystrom or Linformer). Right now, it is unclear exactly what high-frequency signal components are being discarded during the projection.

**W3:** The proposed method implicitly assumes the underlying signal can be well-approximated by a small number of bases $k$. But what happens mathematically if the data is highly non-smooth, containing sharp discontinuities, shocks, or sudden transitions that lack fast spectral decay? Since the functional correspondence relies on these reduced bases, a small $k$ might fail to capture highly localized features due to the uncertainty principle. While the paper tests on some PDEs that can exhibit shocks, it lacks a formal analysis of how the functional mapping behaves, or breaks down, when the underlying function fundamentally violates smooth spatial assumptions.

**W4:** The closed-form solution for the transport operator $C$ relies on inverting a matrix regularized by a parameter $\lambda$. Since the learned bases use a simple softmax and explicitly do not enforce strict orthogonality, the learned basis vectors could easily become highly collinear. Does this rank-collapse cause the unregularized covariance matrix to become poorly conditioned? The paper does not discuss how sensitive the model's numerical stability is to the choice of the Tikhonov regularization parameter $\lambda$, or how the authors would prevent the inversion from diverging during backpropagation when conditioning is poor. Overall, the system lacks a good numerical robustness analysis in the formal sense.

---

> ### Author Rebuttal · Authors · 2026-03-30
>
> We thank qVXE for finding our idea and our results "very interesting".
> # Q1 Theoretical Convergence
> > the bases converge to the span of known optimal bases?
>
> As our bases are learned adaptively through eq. (9), we do not claim convergence to a specific analytic basis, in contrast to [1].
>
> > theoretical sketch of FUNCATTN? approximation error bound?
>
> An intuition is exhibited by decomposing the true operator via:
> $$\lVert\mathcal{T} - \widehat{\mathcal{T}}_m\rVert \le \lVert\mathcal{T} - \mathcal{T}_m^\star\rVert + \lVert\mathcal{T}_m^\star - \widehat{\mathcal{T}}_m\rVert,$$
> where the first term is the best rank-$m$ truncation error (well-known explicit bound via Eckart-Young-Mirsky theorem and polynomially decaying singular values) and the second the learning gap of FuncAttn, that captures the estimation gap induced by the learned basis spans, the closed-form functional alignment, and optimization from data. An explicit bound of this second term is non-trivial and an interesting future direction, we will mention it in our Sec. "Limitations & Future Works".
>
> [1] Convergence of Laplacian Eigenmaps, Belkin & Niyogi
>
> # Q2 Truncation
> > is there a theoretical bound on the truncation error introduced by this implicit low-rank constraint?
>
> See Q3, paragraph Q2
>
> # Q3 Numerical Robustness, Matrix Conditioning
> > Lipschitz continuity under bounded perturbations?
>
> We complement our theoretical analysis (Appendix A) by adding a proposition about the local Lipschitz continuity, following [2, proof of Thm 2]. For $X \in \mathbb{R}^{n\times d}$ with $\lVert X\rVert_2 \le B$,
>
> $Q=XW_q, \quad K=XW_k, \quad V=XW_v,$ with projection
> $\widetilde{Q}=\Phi(X)^\top Q,  \widetilde{K}=\Psi(X)^\top K, \widetilde{V}=\Psi(X)^\top V.$
> Consider the Functional Attention layer
> $$\mathcal{F}(X) = \widetilde{Q}\widetilde{K}^\top (\widetilde{K}\widetilde{K}^\top+\lambda I)^{-1} \widetilde{V}, \quad \lambda>0.$$
> First, the softmax bases are Lipschitz in $X$ with constants $L_{\mathrm{sm},\phi} \le \sqrt{n} \lVert W_{\phi}\rVert$, $L_{\mathrm{sm},\psi} \le \sqrt{n} \lVert W_{\psi}\rVert$ (see [3], Proposition 4).
> Differentiating the projected quantities gives
> $d\widetilde{Q} = d(\Phi^\top)Q + \Phi^\top dQ,$ with $dQ=\Delta X W_q$.
> By submultiplicativity,
> $$\lVert d\widetilde{Q}\rVert_F \le \lVert d\Phi\rVert_F \lVert Q\rVert_2 + \lVert\Phi\rVert_2 \lVert W_q\rVert_2 \lVert\Delta X\rVert_F,$$ and similarly for $d\widetilde{K}$ and $d\widetilde{V}$, each bounded by $O((B L_{\mathrm{sm}} \lVert W_\mathrm{bas}\rVert \lVert W_\mathrm{i}\rVert + \sqrt{n}\lVert W_\mathrm{i}\rVert)\lVert\Delta X\rVert)$ with $i=q,k,v$ and $\mathrm{bas}=\psi,\phi$.
> Now define $$\widetilde{S}:=\widetilde{K}\widetilde{K}^\top+\lambda I.$$ Since $\widetilde{K}\widetilde{K}^\top\succeq 0$ and $\lambda>0$, we have $$\lVert\widetilde{S}^{-1}\rVert_2\le \frac{1}{\lambda}.$$ The layer differential is $$d\mathcal{F} ={} d\widetilde{Q}\widetilde{K}^\top\widetilde{S}^{-1}\widetilde{V} + \widetilde{Q} d\widetilde{K}^\top\widetilde{S}^{-1}\widetilde{V} + \widetilde{Q}\widetilde{K}^\top\widetilde{S}^{-1}d\widetilde{V} - \widetilde{Q}\widetilde{K}^\top\widetilde{S}^{-1}
> \bigl(d\widetilde{K}\widetilde{K}^\top + \widetilde{K} d\widetilde{K}^\top\bigr) \widetilde{S}^{-1}\widetilde{V}.$$
> It follows: $$\lVert d\mathcal{F}\rVert \le \Bigl(\frac{C_1}{\lambda} + \frac{C_2}{\lambda^2}\Bigr) \lVert\Delta X\rVert,$$ where $C_1, C_2$ depend polynomially on $B, n, \lVert W_q\rVert, \lVert W_k\rVert, \lVert W_v\rVert, \lVert W_\phi\rVert, \lVert W_\psi\rVert$. That concludes the proof of the local Lipschitz continuity and gives an explicit bound.
>
> > Q2 Truncation Error
>
> It follows that if the $m$-dimensional basis bottleneck is viewed as replacing a non-compressed $X$ by its projection $X_{m}=P_{m}​X$ then $\lVert \mathcal{F}(X)−\mathcal{F}(X_{m})\rVert \le L \lVert X−X_{m} \rVert$. Error induced by the low-rank truncation is directly governed by the projection of the input in the learned basis, that also gives an interpretation for the absorption of shocks.
>
> > behavior of the condition number during training? How sensitive is the stability and performance to λ?
>
> We parameterized $\lambda = \text{nn.sigmoid}(\alpha)$ where alpha is a learnable scalar. We monitor the average condition number over 8 layers of the inverse matrix in Eq 7 (http://anonymous.4open.science/r/Figures-8D4F/condition_number.png), and we observe that with decreasing $\alpha_{init}$, the condition number goes up. The table reports the performance on Elasticity, showing that our model is robust to initialization; relaxing regularization yields a mild performance gain.
> | $\alpha_{\text{init}}$ | $\kappa$ (final) | Test Error |
> |:-:|:-:|:-:|
> | 0  | ~100 | **0.48** |
> | 3  | ~90  | 0.49 |
> | 6  | ~8   | 0.50 |
>
> [2] SpecFormer: Guarding Vision Transformer Robustness via Maximum Singular Value Penalization, Hu et al
>
> [3] On the Properties of the Softmax Function with Application in Game Theory and Reinforcement Learning, Gao & Pavel

---

> > ### Author Rebuttal · Reviewer_qVXE · 2026-04-03
> >
> > Thanks for the response. I will be maintaining my score, provided the Lipschitz proof and condition number analyses are integrated into the final manuscript.

---

> > > ### Author Response · Authors · 2026-04-08
> > >
> > > We greatly appreciate your engagement throughout the discussion and are glad that our responses addressed your concerns. The Lipschitz proof, condition number analyses, and related clarifications will be carefully incorporated into the final manuscript, as outlined in the rebuttal. Thank you again for helping us improve the paper.

---

### Official Review · Reviewer_5pVm · 2026-03-12

**Soundness:** 3
**Presentation:** 3
**Significance:** 3
**Originality:** 4
**Overall Recommendation:** 4
**Confidence:** 3

**Summary:**

This paper presents a novel construction for an attention-like mechanism by using the Functional Maps framework, which was initially proposed in the shape matching/geometry processing literature. The high-level idea presented by the authors is that instead of building a complete O(N^2) dot product attention matrix, the authors propose to use a low-rank representation of the map, which is inspired by the functional map construction -- i.e., based on treating the given signal as samples from some functional space. To achieve this, the authors propose a Functional Attention oeprator (FuncAttn) given in Eq. (5) of the paper, which, computationally, reduces to estimating a matrix C, whose size is k x k, regardless of the dimensionality N of the input signal (where k << N). The authors complement this idea by introducing a learned adaptive basis, which is meant to provide an efficient (data-dependent) way to encode the signals.

The authors demonstrate the utility of their approach in several applications, ranging from signal regression, PDE solving and RNA segmentation. The proposed approach outperforms the baselines in all of these settings.

**Compliance With Llm Reviewing Policy:**

Affirmed.

**Ethical Review Flag:**

Flag this paper for an ethics review.

**Final Justification:**

I maintain my score. As mentioned during the discussion period, I was somewhat surprised and confused by the inconsistencies between the authors' responses and the statements in the paper (whether what is learned is a basis or a projection operator, the non-injectivity of the functional map representation and potential loss of detail due to basis truncation, the geometric nature of the Tikhonov regularization, etc.). The authors stepped back and tried to clarify these issues in a follow-up discussion, which was helpful. If the paper is accepted, I _strongly_ urge the authors to both provide the a technically rigorous discussion of the points raised in the rebuttal period, and also to clearly acknowledge conceptual limitations (such as using regularization to stabilize training and leaving more principled approaches as future work).

**Key Questions For Authors:**

While I really like the proposed paper, I have some questions related to the specific design choices and possible technical issues.

1. The regularization proposed by the authors is a Frobenius norm (Tikhonov-like) regularization on the matrix C -> Eq (6) of the paper. This is quite different from what is done in the functional maps literature, where standard regularizations involve commutativity with the Laplacian or Orthonormality of the matrix C. See Sections 2.4.4 and 2.4.5 here: https://www.lix.polytechnique.fr/~maks/fmaps_SIG17_course/notes/siggraph17_course_notes.pdf#page=15.10) What is particularly strange is that when lambda is very high, the solution to Eq. (6) is just the zero matrix. This does not seem like a reasonable map, and I feel overall this regularization is not well-motivated, and potentially conceptually unsound. What is the reasoning behind this particular regularization?

2. The formulas in Eq. (4) assume that the basis is orthonormal (and thus its inverse is its transpose). However, when _learning_ the basis in Section 4.2, the basis is unconstrained. Although in Section 5.6 the authors provide empirical evidence that a freely learned basis performs better, I still wonder about the validity of the formulas (using the transpose instead of the pseudo-inverse, for example) for the conceptual framework of the paper. Can the learned matrix truly be considered *a basis* if it is not orthonormal, and we compute coefficients simply by using the transpose?

3. In many applications the attention has to be able to represent an arbitrary permutation matrix over the samples/tokens, does the low-rank decomposition enable (or not) this behavior? Are there limitations to how well the functional attention layer can represent the underlying mapping?

4. Does the requirement of solving the linear system at test time (Eq. (7) of the paper) induce computational overhead at test time, especially for long output sequences? I understand that the matrix inverse is independent of the length of the sequence (and only requires inverting a $k \times k$ matrix). However, for regular attention, e.g., in the auto-regressive setting with multiple heads, we have to compute the attention *many times* at test time (one for each generated token), and I wonder if the matrix inverse can become prohibitive. Moreover, what is the tradeoff between increasing $k$ and the computational complexity? I see that there is some computational analysis in Appendix D. However, it would be interesting and important to understand the applicability of the proposed approach in an autoregressive setting (unless, I'm mistaken, it seems that the authors only test on tasks where the full sequence is processed at once).

5. The authors demonstrate that method tends to generalize across grid sizes in Section 5.5. However, I still wonder to what extent the proposed approach is applicable to an arbitrary sequence length (as this is one of the key advantages of the standard attention mechanism). Does the fact that the basis is learned affect this generalization capability? Have the authors observed situations in which the learned nature of the basis can constrain generalization?

**Limitations:**

Yes, the authors provide a limitations section, acknowledging some weaknesses and suggesting future work.

**Strengths And Weaknesses:**

I think the main strength of this paper is the novelty of the proposed attention operator and the explicit links to the functional maps framework. The authors first make a nice conceptual connection across two different domains, and, second, show that FunctionalAttention operator can be useful in a range of applications. I think this can lead to follow-up work since the functional map literature is quite rich so it's likely that many interesting and fruitful ideas that will arise from this connection.

The main weaknesses are related to the technical soundness of the proposed approach, which are summarized below in the Questions and Limitations sections below.

---

> ### Author Rebuttal · Authors · 2026-03-30
>
> We are delighted that you acknowledge the novelty of our paper and appreciate the “nice conceptual connection” between attention and the functional maps framework, which can lead to “many interesting and fruitful ideas”. Below we address the concerns raised in the review.
> # Q1: Tikhonov vs. Geometric Regularizer in Classical Shape Matching
> While Tikhonov regularization is a standard approach in many machine learning domains, its history in classical shape matching is more nuanced. As noted by [Ovsjanikov et al, SIGGRAPH Course Notes, 2017, p.35], early applications in that field yielded suboptimal results, prompting a shift toward more complex, geometrically inspired regularizers.
>
> However, our framework introduces a distinct setting: the functional map C is estimated on-the-fly from Q and K within a learned spectral space characterized by data-dependent, adaptive bases (Eq. 9). Adhering to the principle of Occam’s Razor, we initially adopted Tikhonov regularization, which has proven empirically robust in our experiments. While we acknowledge that incorporating more sophisticated geometric regularization represents a promising direction for future research, the current formulation provides a stable and effective baseline.
>
> Regarding the implementation of Eq. 6, we parameterize the regularization weight as $\lambda=\text{nn.sigmoid}(\alpha)$, effectively bounding $\lambda \in (0,1)$. This parameterization, combined with the Tikhonov term, ensures positive definiteness of the system matrix and prevents the numerical instability associated with $\lambda \to \infty$. Furthermore, we provide an ablation study on the condition number of the matrix in Eq. 7 to demonstrate the numerical health of the linear solve (see qVXE Q3 for details).
> # Q2: Non-Orthonormal Basis
> We appreciate this insightful observation. The learned operator $\mathbf{\Phi}^\top$ in Eq. 4 is specifically designed for direct latent projection to compute the spectral coefficients $\tilde{\mathbf{Q}} = \mathbf{\Phi}^\top \mathbf{Q}$; it is not intended to serve as a numerical approximation of the inverse of $\mathbf{\Phi}$.
>
> Furthermore, for a generalized functional basis, orthonormality is not a strict requirement, a property consistent with observations in Marin et al. NeurIPS 2020. We have addressed this distinction in Remark 4.2 and Appendix A4, and we will further refine these sections in the final manuscript.
> # Q3: Low-Rank Expressivity.
> While a functional map $\mathbf{C}$ possesses a lower rank than a permutation matrix $\mathbf{P}$, it encodes a broader class of functional correspondences. This representation strictly subsumes the space of permutation matrices; specifically, while every $\mathbf{P}$ has an associated correspondence $\mathbf{C}$, the converse is not true (Ren et al., CFG 2021). Furthermore, functional maps facilitate a larger matching space by enabling 'soft' permutations—relaxing entries from $\{0,1\}$ to $[0,1]$—which is inherently more compatible with the continuous nature of attention mechanisms. Additionally, we provide a comparative discussion with other low-rank attention methods in Q1 of BfSd.
> # Q4: Efficiency of Linear Solve Eq.7 During Testing and its Behavior with Increasing k.
> Via the Woodbury identity (Appendix D), the system reduces from $k \times k$ to $d_{\text{head}} \times d_{\text{head}}$. Since $d_{\text{head}}$ is fixed (typically 16), solve cost is constant w.r.t. both $n$ and $k$.
> Empirically, sweeping $k \in \{32, 64, 128, 256\}$ on an RTX A2000 ($B{=}2, N{=}861, C{=}128, k{=}64， h{=}8$) shows:  the solve time stays flat at ~0.36 ms and makes only ~26% of the total computation budget, while only basis projection cost grows. The **total FuncAttn is 0.76× standard MHA**.
>
> Experiments in our paper process full discretizations at once, which is standard in operator learning. The autoregressive setting is not our focus and left for future research.
> # Q5: Generalization to Arbitrary Sequence Length and Failure.
> FuncAttn natively supports arbitrary sequence lengths: (1) The learned bases are continuous functions of spatial coordinates via $B = \text{Softmax}(\text{Linear}(X))$ (Eq. 9), so for any new $x \in \Omega$ the basis value is well-defined (Proposition 4.1), (2) $C \in \mathbb{R}^{k \times k}$ is independent of $n$. Thm A.3 further establishes FuncAttn as a learnable integral operator, guaranteeing discretization invariance.
>
> The learned basis helps generalization: fixed bases assume regular grids, while ours adapts to each input. Tab 6 shows that learned bases outperform Fourier. Sec 5.4 shows FuncAttn achieves the best OOD generalization under unseen Reynolds numbers.
> While under extreme distribution shift, FuncAttn may degrade, this is a shared limitation of all data-driven methods.

---

> > ### Author Rebuttal · Reviewer_5pVm · 2026-04-03
> >
> > I thank the authors for their thoughtful responses. Unfortunately several of my questions are still not completely answered.
> >
> > 1. Tikhonov regularization: the main question that I still have is whether this regularization actually promotes a meaningful structure or whether it its role is primarily to stabilize training? The question of the behavior as $\lambda$ tends to infinity is not simply one of instability. Rather, it highlights that as $\lambda$ grows, the regularization actually promotes _undesirable_ properties of the functional map ($C -> 0$). Stating that this is just an training-oriented regularization is perfectly fine for me, but I think the authors should avoid attributing structural role to this regularizer if such a role doesn't exist.
> >
> > 2. The rebuttal mentioned. "The learned operator $\mathbf{\Phi}^\top$ in Eq. 4 is specifically designed for direct latent projection to compute the spectral coefficients". This directly contradicts the core claims in the paper that $\mathbf{\Phi}$ is meant to be a learned _basis_. This is mentioned many times in the paper, including e.g., in the abstract "We introduce Functional Attention, which reinterprets attention as a functional correspondence between adaptive bases." or Section 2: "While we also adopt this functional view, we distinguish our work by explicitly learning a set of bases", etc.) Please note that $\mathbf{\Phi}$ cannot be both a basis and a projection operator at the same time, unless the basis is orthonormal. The claim in the rebuttal is completely different from what is claimed in the paper.
> >
> > I also thank the authors for mentioning [Marin et al. 2020] paper. After looking at that paper, I see a fundamental difference: while  [Marin et al. 2020] also use an unconstrained learned basis, they compute the _pseudo-inverse_ of the learned basis (see Section 4.1 Learning a linearly-invariant embedding of that paper), and not the transpose (see "this can be computed by _solving a linear system of equations_. Importantly, this procedure can be differentiated using the closed-form expression of derivatives of matrix inverses...") Again, this is fundamentally different from what is done in this paper.
> >
> > 3. Low-Rank Expressivity. The authors mention in the rebuttal, "This representation strictly subsumes the space of permutation matrices; specifically, while every $\mathbf{P}$ has an associated correspondence $\mathbf{C}$, the converse is not true (Ren et al., CFG 2021)." Unfortunately, this claim is false. in fact, while every $\mathbf{P}$ has an associated correspondence $\mathbf{C}$, two distinct $\mathbf{P}$ might be associated with the same (or numerically very close) $\mathbf{C}$. Imagine, for example, the case, of $k=1$ where we only have 1x1 functional maps. Regardless of the point-to-point map, all functional maps will be the same. This relation between "map compression" and the dimensionality of the functional maps was analyzed and exploited in Ren et al. "MapTree: Recovering Multiple Solutions in the Space of Maps" Proc. SIGGRAPH Asia, 2020. The authors did not address the question of how the low-rank approximation (and thus smoothing) of the space of maps affects the performance of their method.
> >
> > 4. Thank you. I understand that the auto-regressive application is outside of scope of this work.
> >
> > 5. Q5: Thank you. Perhaps I missed it, but it seems that the _generalization_ across $N$ is only empirically demostrated in one case: Table 4 for 1D Burgers’ equation. Have you tested this zero-shot generalization across varying $N$ on complex, irregular 2D or 3D topologies (such as the Elasticity or RNA segmentation tasks)?
> >
> > Overall, I remain positive about this work. However, after reading the rebuttal, which introduces several technical inaccuracies, I'm less positive than before. I do believe that the limitations and lack of conceptual justification for different parts of the paper have to be acknowledged clearly, to both delineate the contribution and enable follow-up work.

---

> > > ### Author Response · Authors · 2026-04-07
> > >
> > > We thank the reviewer for the careful follow-up and insightful questions.
> > > 1. We acknowledge that the Tikhonov regularization in eq. 6 & 7 is primarily for stabilization purposes, as it has been initially designed for (see e.g. Tikhonov Regularization and Total Least Squares, Golub et al., 1999). As we implement $\lambda = \text{nn.sigmoid}(\alpha) \in (0,1)$, the undesirable properties of $\lambda \rightarrow \infty$ does not happen. When removing Tikhonov, namely when the stationary condition of KKT is $\mathbf{C} \tilde{K} \tilde{K}^T = \tilde{Q} \tilde{K}^T$, the training process breaks (model outputs NaN), which necessitates the employment of Tikhonov. We will clarify this point in the final manuscript.
> > > To quantify the effect of stabilizing training using Tikhonov, in Fig. (http://anonymous.4open.science/r/Figures-8D4F/condition_number.png), we show that when increasing the initial $\alpha_{init}$ from 0 to 6, i.e. increasing the initial regularization effect, the final condition number decreases from 100 to 8, hence strengthening the stable training behavior.
> > >
> > >
> > > 2. Thanks for your clarification. We acknowledge that the basis and projection matrices are normally different and the projection matrix is the transposed basis only when the basis matrix is orthonormal. Theoretically, a pseudo-inverse $\tilde{\mathbf{Q}} = \Phi^\dagger \mathbf{Q}$ should be employed in eq. 4, where $\Phi^\dagger = (\Phi^T \Phi)^{-1} \Phi^T$. However it leads to exploding gradients, which breaks the training pipeline badly in our experiments.
> > > To overcome this issue, we implemented a *stabilized pseudo inverse* by using Tikhonov $(\Phi^T\Phi + \lambda I)^{-1}\Phi^T$ (a classical approach, see e.g. eq. (2.1) in Ridge Regression: Biased Estimation for Nonorthogonal Problems, Hoerl and Kennard, 1970), which leads to good results (0.0051 test error on elasticity, 0.0044 test error on darcy). However, in practice, we found that using the transpose instead of the stabilized pseudo inverse leads to similar results (0.0050 test error on elasticity, 0.0042 test error on darcy), while being more computationally efficient and getting rid of the stability issue of the vanilla pseudo inverse in one go.
> > > Another practical consideration is to solve the convex problem eq. 6 & 7, where a matrix inversion has to be computed. The condition number of the matrix to be inverted plays a key role in the training stability as well.
> > > We report in (https://anonymous.4open.science/r/Figure2-298D/elasticity_condition_number.png) the condition number of $\tilde{K} \tilde{K}^T  + \lambda I_k$ of the stabilized pseudo-inverse (yellow) and transpose (purple), and it shows that the simple transpose leads to better condition numbers, hence stable linear solve to compute the functional maps in eq. 7, which contributes to the performance and robustness of our whole pipeline. We will explicit these discussions in the camera-ready paper, that will only require minor changes.
> > >
> > >
> > > 3. Thank you for this important question, we acknowledge that our previous phrasing was misleading. We agree with the reviewer on the following important point: the functional map representation is not lossless for arbitrary low rank $k$, and its expressivity depends directly on the number of chosen bases. In our method, this truncation should indeed be interpreted as a form of spectral smoothing, while larger $k$ preserves finer correspondence details.
> > > Finally, let us highlight that we do study the practical effect of this dimensionality reduction in our experiments by varying $k$ (see Tab 5 & 10 in our paper). Also, the response to Reviewer 4Cdf, Q2, complements this analysis and will be added to the camera-ready version.
> > >
> > >
> > > 5. Q5: Thank you for this question, we conducted new zero-shot generalization experiments on the Elasticity benchmark, that we will be happy to add in our paper to further highlight the advantage of our method concerning its generalization. Following the data generation protocol of GeoFNO (Li et al., 2024), we generated ~5000 points/samples and resampled them to resolutions of 500, 1000, 1500, 2500, 3500, 4000, and 5000. All models were **trained exclusively at resolution 500** and evaluated zero-shot on all other resolutions. We compare FuncAttn against Transolver and Galerkin Transformer under identical settings, and our method consistently outperforms its competitors.
> > > | Resolution     | Transolver | Galerkin | FuncAttn (Ours)  |
> > > |:--------------:|:----------:|:--------:|:----------------:|
> > > | 1,000          | 0.1338     | 0.1583   | **0.1191**       |
> > > | 1,500          | 0.1379     | 0.1644   | **0.1192**       |
> > > | 2,500          | 0.1439     | 0.1769   | **0.1229**       |
> > > | 3,500          | 0.1519     | 0.1853   | **0.1284**       |
> > > | 4,000          | 0.1544     | 0.1864   | **0.1315**       |
> > > | 5,000          | 0.1556     | 0.1875   | **0.1320**       |
> > >
> > > See visualizations in (https://anonymous.4open.science/r/Figure2-298D/cross_resolution_comparison.png).

---

### Decision · Program_Chairs · 2026-04-30

**Decision:**

Accept (regular)

**Comment:**

All reviewers acknowledged the novelty of reinterpreting attention as a functional correspondence via the functional maps framework, and the extensive experiments across PDEs, 3D segmentation, and regression consistently demonstrate strong performance and resolution invariance.

The primary concerns — lack of formal theoretical guarantees (qVXE), conceptual inconsistencies around Tikhonov regularization and the basis/projection distinction (5pVm), and originality relative to low-rank operators (BfSd) — were largely addressed through thorough rebuttals.

For the camera-ready, the authors should rigorously clarify the stabilization-only role of Tikhonov regularization, the basis vs. projection distinction, and integrate the promised Lipschitz and conditioning analyses, as urged by Reviewer 5pVm.